# Learning Multiscale Non-stationary Causal Structures

**Gabriele D'Acunto**                                                                 *gabriele.dacunto@uniroma1.it*
*DIAG, Sapienza University of Rome*
*Centai Institute, Turin, Italy*

**Gianmarco De Francisci Morales**                                                              *gdfm@centai.eu*
*Centai Institute, Turin, Italy*

**Paolo Bajardi**                                                                        *paolo.bajardi@centai.eu*
*Centai Institute, Turin, Italy*

**Francesco Bonchi**                                                                   *francesco.bonchi@centai.eu*
*Centai Institute, Turin, Italy*

**Reviewed on OpenReview:** *https://openreview.net/forum?id=SQnPE63jtA*

## Abstract

This paper addresses a gap in the current state of the art by providing a solution for modeling causal relationships that evolve over time and occur at different time scales. Specifically, we introduce the *multiscale non-stationary directed acyclic graph* (MN-DAG), a framework for modeling multivariate time series data. Our contribution is twofold. Firstly, we expose a probabilistic generative model by leveraging results from spectral and causality theories. Our model allows sampling an MN-DAG according to user-specified priors on the time-dependence and multiscale properties of the causal graph. Secondly, we devise a Bayesian method named *Multiscale Non-stationary Causal Structure Learner* (MN-CASTLE) that uses *stochastic variational inference* to estimate MN-DAGs. The method also exploits information from the local partial correlation between time series over different time resolutions. The data generated from an MN-DAG reproduces well-known features of time series in different domains, such as volatility clustering and serial correlation. Additionally, we show the superior performance of MN-CASTLE on synthetic data with different multiscale and non-stationary properties compared to baseline models. Finally, we apply MN-CASTLE to identify the drivers of the natural gas prices in the US market. Causal relationships have strengthened during the COVID-19 outbreak and the Russian invasion of Ukraine, a fact that baseline methods fail to capture. MN-CASTLE identifies the causal impact of critical economic drivers on natural gas prices, such as seasonal factors, economic uncertainty, oil prices, and gas storage deviations.

## 1 Introduction

A causal graph describes causal relationships among the constituents of a given system, and represents a powerful tool to analyze such a system under interventions and distribution changes. In general, causal graphs are unknown. Fortunately, it is possible to leverage causal structure learning approaches to unveil and quantify the causal relationships among variables. While randomized experiments are the gold standard for testing causal hypotheses (especially in medicine and the social sciences), in many cases such interventional approaches are unfeasible or unethical. Hence, great effort has been devoted to the development of methods able to retrieve causal structures from observational data (Glymour et al., 2019; Schölkopf et al., 2021).

Regardless of the different causal structure learning methods, the most informative causal graph is a *directed acyclic graph* (DAG), where the nodes in $\mathcal{V}$ are the variables of the system, all possible edges $e_{ij} \in \mathcal{E} \subseteq \mathcal{V} \times \mathcal{V}$ are directed and represent direct causal effects, and feedback loops among nodes are forbidden (acyclicity requirement). A DAG can be associated with its functional representation, also known as *structural equation model* (SEM, Pearl 2009). Here each node of the causal graph is written as a function of the values of a set of parents nodes and of an endogenous latent noise (see Appendix A). In this work, we focus on the case in which such functions are linear and the latent noise is additive.

Even though widely studied and applied, a linear SEM is not adequate to cope with causal relations that evolve over time and occur at different time scales, which are both common when dealing with time series. Indeed, a SEM assumes that (i) causal edges and their weights are stationary and (ii) there is only one time scale at which causal relations occur, i.e., the one associated with the frequency of observed data. However, in practice, causal structures might be non-stationary (Zhang et al., 2021b; Raggad, 2021; D'Acunto et al., 2021) and often there is no prior knowledge about the temporal resolutions at which causal relations occur (Besserve et al., 2010; Gong et al., 2015; Runge et al., 2019; D'Acunto et al., 2022).

To overcome these limits, we introduce multiscale non-stationary causal structures, namely MN-DAGs, that generalize linear DAGs to the time-frequency domain. In our work, the term *multiscale* means that we consider multiple time resolutions, i.e., frequency bands. Hence, we look for causal interactions among time series within each of those distinct frequency bands, and we simultaneously inspect the behaviour of these causal relationships along time. Throughout the paper, we use $2^j$ to represent a certain temporal resolution, where $j = \{1, \ldots, J\}$ indicates the associated scale level and $J \in \mathbb{N}$ is the maximum level considered. To clarify the meaning of time scale, let us consider a data set $\mathbf{X} \in \mathbb{R}^{N \times T}$ made by $N$ time series of length $T = 2^J$. The column $\mathbf{X}[t] = [X_1[t], \ldots, X_N[t]]'$ represents a sample collected at frequency $\Delta t$. For example, let us say that the time series are observed at daily frequency, hence $\Delta t$ is equal to one day. The scale level $j = 1$ refers to the variations of the time series associated with the time scale of $2^1 \Delta t$, i.e., two consecutive days. Analogously, the scale level $j = 2$ refers to the variations of the time series associated with the temporal resolution of $2^2 \Delta t$, i.e., four consecutive days. And so on and so forth, until we reach the maximum level $j = J$. Additionally, the $j$-th time scale corresponds to the frequency band $[1/2^{j+1}, 1/2^j]$.

In MN-DAGs each time scale is represented by a different graph page (akin to multi-layer networks). Then, the vertices within a certain page are associated with the multiscale representation of the $N$ time series at the frequency band corresponding to that page. There exists a unique global causal ordering '$\prec$' shared by all graph pages, such that the possible parent set $\mathcal{P}_{i,\prec}$ for the $i$-th node $X_i$ can include only those nodes $X_j$ that precede it in the causal ordering ($X_j \prec X_i$). Causal relationships among nodes, represented as directed edges, can vary smoothly over time and constitute acyclic structures within each time scale. So, throughout the paper, the term non-stationarity associated with causal structures refers to a smooth dependence on time, similarly to how it is defined by Huang et al. (2020).

To achieve our goal, the main technical challenges we face are: *(i)* the definition of a probabilistic generative model that allows to sample an MN-DAG; *(ii)* the development of a learning method to estimate MN-DAGs from real-world data. Regarding the first point, we propose a probabilistic generative model over MN-DAGs having the causal ordering and the causal relationships as latent variables. The observables are $N$ zero-mean time series of length $T$. Our model leverages linear SEM and multivariate locally stationary processes (MLSW, Park et al. 2014). In particular, MLSW is a mathematical framework to represent time series as a sum of contributions coming from different time scales (see Appendix C). Concerning the second point, we propose a Bayesian method, called MN-CASTLE, that uses stochastic variational inference (SVI, see Appendix D) for learning causal structures. MN-CASTLE is able to cope with multiscale data that features time-dependent variance. Our method relies upon observational data and the estimate of the inverse of the power spectrum at different temporal resolutions.

Overall, our contributions can be summarized as follows:

- We define a new type of causal structure for modeling causal relationships that evolve over time and occur at different time scales (MN-DAG).

- We devise a probabilistic generative model that allows sampling an MN-DAG according to user-specified priors on the time-dependence and multiscale properties of the domain. Our model can be used to generate synthetic time series with real-world characteristics.

- We design a Bayesian inference method, MN-CASTLE, for estimating MN-DAGs from real-world data. Our empirical assessment on synthetic datasets demonstrates that MN-CASTLE outperforms baseline methods in various experimental settings and is robust to model misspecification.

- When applied to study what drives natural gas prices in the US market, MN-CASTLE succeeds where baselines fail. In fact, it is the only method able to capture the dynamic nature of the market and the impact of exogenous events such as COVID-19 and the Russian invasion of Ukraine.

At a high level, *this paper bridges the gap between multiscale modeling and machine learning-based causal structure learning methods.*

**Roadmap.** This article is organized as follows. Section 2 relates our proposals to existing methods, highlighting differences and similarities. Then, the subsequent three sections deal with our first technical challenge. Specifically, Section 3 presents our probabilistic generative model, Section 4 details how an MN-DAG is sampled in our model, and Section 5 how to generate data from the sampled MN-DAG. At this point, we address our second challenge in Section 6, where we pose a Bayesian learning method developed according to the proposed probabilistic generative model. Next, Section 7 presents the empirical assessment of our model. In detail, Section 7.1 statistically describes data generated by the probabilistic generative model. Section 7.2 regards tests on synthetic datasets, by providing details concerning the experimental settings and introducing the considered baseline models. Subsequently, Section 8 analyses a real-world use case on natural gas prices in the US market. Finally, Section 9 concludes with an additional discussion concerning our findings, and outlines open questions and future research directions.

## 2 Related Work

Causal structure learning methods can be mainly classified into three categories, according to the approach used to infer the causal graph: (i) *constraint-based approaches*, which make use of conditional independence tests to establish the presence of a link between two variables (Spirtes et al., 2000; Huang et al., 2020); (ii) *score-based methods*, which use search procedures in order to optimize a certain score function (Heckerman et al., 1995; Chickering, 2002; Huang et al., 2018); (iii) *functional causal models*, which express a variable at a certain node as a function of its parents (Shimizu et al., 2006; Hoyer et al., 2008; Zhang & Hyvärinen, 2010; Shimizu et al., 2011; Peters et al., 2014; Bühlmann et al., 2014). Our approach fits into the latter category and aims to handle the presence of non-stationarity and different temporal resolutions in the underlying causal structure.

Unlike the multiscale causal structure learning method proposed by D'Acunto et al. (2022), which estimates multiscale stationary causal relationships hinging on stationary wavelet transform (Nason & Silverman, 1995) and non-convex optimization, our method applies a different learning scheme and is able to handle non-stationary relationships as well. Furthermore, the method we propose exploits the estimate of the decomposition of the inverse power spectrum at different time scales, whereas the algorithm proposed in the previous paper operates on the estimated wavelet detail vectors.

In the past, several approaches have been developed that can infer causal structures in the presence of non-stationarity under certain assumptions (Song et al., 2009; Ghassami et al., 2018; Strobl, 2019; Perry et al., 2022). The main (implicit) assumption common to these approaches, concerns the time scale at which causal interactions occur, that is, it is assumed that this scale coincides with the frequency of observation of the data. The model we propose, relaxes this assumption, and allows time-dependent causal relationships to be investigated at different temporal resolutions. Another difference concerns the assumption regarding the existence of multiple domains, where causal dependencies between variables may vary but are assumed to be stationary within each domain, to exploit non-stationarity and distributional shifts to recover the underlying causal structure. Although in the context of time series, the dataset can be segmented into different domains through a sliding window approach, this procedure introduces discretionary choices such as (i) the choice of the splitting points and (ii) the size of the time window in which causal relationships should be stationary.

However, in general, for real data there is no prior knowledge regarding the above issues: the causal structure might vary a lot even when windows are overlapping (D'Acunto et al., 2021). In contrast, our method aims to learn the causal structure and describe its temporal evolution, assuming that it is linear in the frequency domain and that the causal ordering is shared between the temporal resolutions considered.

Our probabilistic generative model extends the works of Cundy et al. (2021); Charpentier et al. (2022), since it is suitable for time-series data and provides a causal structure that lives in the time-frequency domain. Even though our approach leverage Gumbel distributed variables for sampling the causal ordering as in the previous two works, the procedure we apply is different and requires a lower computational cost (Gadetsky et al., 2020). In addition, our inference model uses a gradient estimator with a data-dependent control variate strategy for learning the parameters of the causal ordering distribution, whereas existing models exploit differentiable relaxations of such a distribution. Our procedure uses the masking of distributions as well to optimize at each step only the causal relations compliant to a certain causal ordering. A similar approach is also employed in Ke et al. (2019); Ng et al. (2022). However the masking used in those works aims at excluding all non-causal relations, not just those that do not conform to the causal ordering.

Finally, we exploit recent developments in variational inference in order to approximate the posterior distribution over MN-DAG parameters given data, in accordance with the MN-CASTLE probabilistic model. This general learning scheme is also exploited in other recent works (Cundy et al., 2021; Charpentier et al., 2022; Annadani et al., 2021; Lorch et al., 2022) to model the posterior distribution over the parameters of a DAG, as defined in the corresponding proposed probabilistic models.

## 3 Probabilistic Generative Model

The probabilistic generative model over MN-DAGs we put forth incorporates both the causal ordering and the causal relationships as latent variables. The observables consists of $N$ zero-mean time series, each of length $T$. In our model, the underlying causal structure determines the *transfer function matrix*. Specifically, this matrix is defined as a time-dependent mixing matrix that depends on the causal structure and the strength of causal relations at each time step and scale level. Furthermore, the transfer function matrix provides a measure of the local variance and cross-covariance between the time series, which is equivalent to the power spectral matrix (see Section 5). From a linear conditional dependence perspective, the inverse of the power spectral density is a concept extensively studied in spectral theory (Dahlhaus, 2000, and references therein). Indeed, the inverse of the power spectral density is a generalization of the precision matrix to the time-frequency domain. It provides information about the local linear dependence between two time series after removing the linear effects of the rest of the time series. Therefore, according to our model both the power spectral matrix and its inverse are driven by the multiscale time-dependent causal structure. This dependence on the causal structure implies that the power spectral matrix and its inverse are also time-dependent.

The proposed probabilistic model takes as input (i) the number of nodes $N \in \mathbb{N}$; (ii) the number of samples $T \in \mathbb{N}$; (iii) a parameter $\mu \in [0,1]$ associated with the multiscale feature; (iv) $\tau \in [0,1]$ which describes the time dependence of causal relationships; (v) $\delta \in [0,1]$ that manages the density of the MN-DAG.

Figure 1 shows a $(\tau, \mu)$-quadrant along with examples of latent causal structures which determine the sampled data, according to the specified values of $\mu$ and $\tau$. When $\mu = 0$, we obtain the single-scale case depicted in Figure 1(a) and 1(b). Here, the MN-DAG has only one page. We assume that the power spectrum that describes the system is concentrated in the finest scale level $j = 1$. In addition, if also $\tau = 0$, the causal links are stationary (Figure 1(a)). Starting from the origin, as we move to the right ($\tau \to 1$) the temporal dependence of causal connections increases. As we move upwards ($\mu \to 1$), the likelihood that the causal graph contains more pages increases (Figure 1(c) and 1(d)). Then, the overall power spectrum is spread over more temporal resolutions.

The following Sections 4 and 5 delve into the sampling of the MN-DAG and the generation of data, respectively.

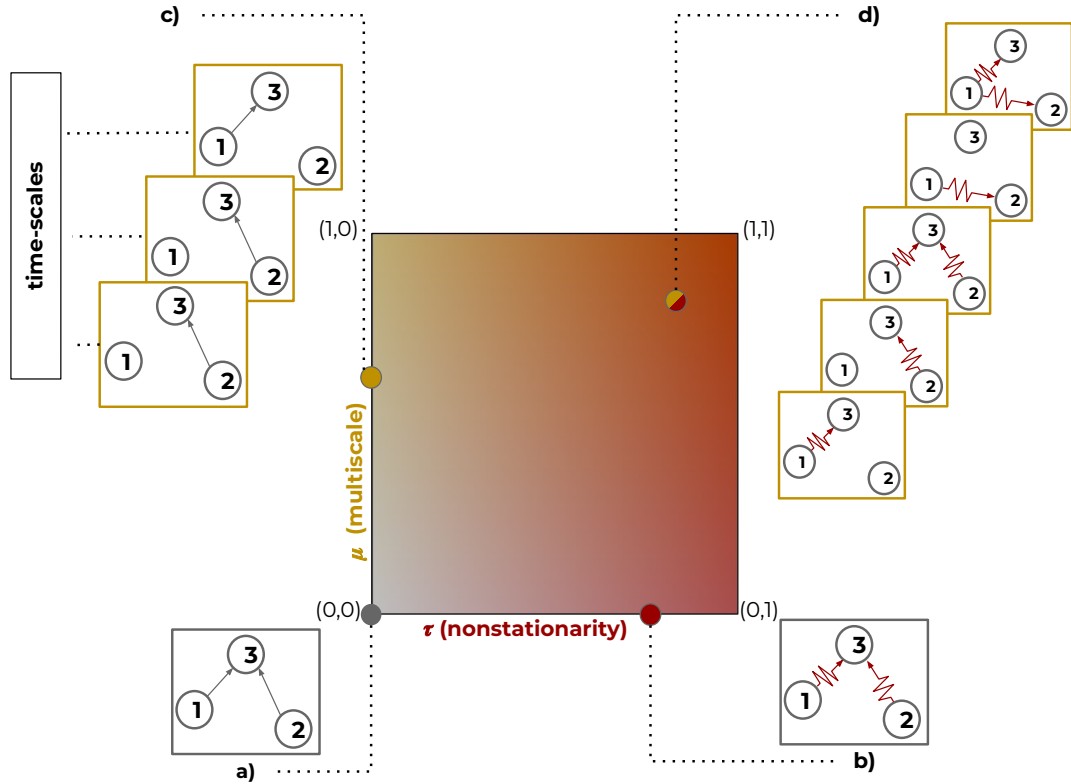

Figure 1: Examples of causal structures that can be sampled from the proposed probabilistic model according to the specified values for non-stationarity and multiscale features. In the depicted quadrant, we have the non-stationarity feature (associated with the parameter $\tau \in [0,1]$) on the x-axis in red, and the multiscale feature (associated with the parameter $\mu \in [0,1]$) on the y-axis in yellow, respectively. Colors, edges shape and number of graph layers highlight differences from the single-scale stationary DAG corresponding to the origin of the quadrant (a). When we move horizontally, the temporal dependence of the causal coefficients (edges in the causal graph) changes (b). Similarly, vertical shifts in the quadrant are associated to the change in the number of time scales (pages of the causal graph) contributing to the sampled data (c). Finally, when both $\tau$ and $\mu$ are different from zero, we sample data concerning a system driven by an underlying multiscale non-stationary causal structure (MN-DAG).

## 4 Sampling an MN-DAG

Figure 2 shows the three steps needed to sample an MN-DAG.

**Sample the time scales.** The number of pages (time scales) of the MN-DAG is $J = 1 + J'$, where $J'$ is sampled from a binomial distribution $J' \sim B(\log_2(T) - 1, \mu)$. Here, the first parameter of the binomial distribution is the number of trials and $\mu$ represents the probability of success. Without loss of generality, we assume that temporal resolutions are consecutive, i.e., given the value of $J$, all the time scales $2^j$, $j = \{1, \ldots, J\}$, are associated with a page in the causal graph. This assumption does not imply that causal relations occur within all the considered pages. Since the model is probabilistic and the user specifies a value for the density $\delta$, we also might end up with a causal graph without edges.

**Sample the causal ordering.** Within our probabilistic model, we assume that the causal ordering $\prec$ is shared by all time scales. This property implies that, given $\prec$, the possible parent sets at each temporal resolutions $\mathcal{P}_{i,\prec}$ for the $i$-th variable $X_i$ are $\{\mathcal{P}_i \mid \mathcal{P}_i \prec X_i\}$. The causal ordering $\prec$ can be thought as a

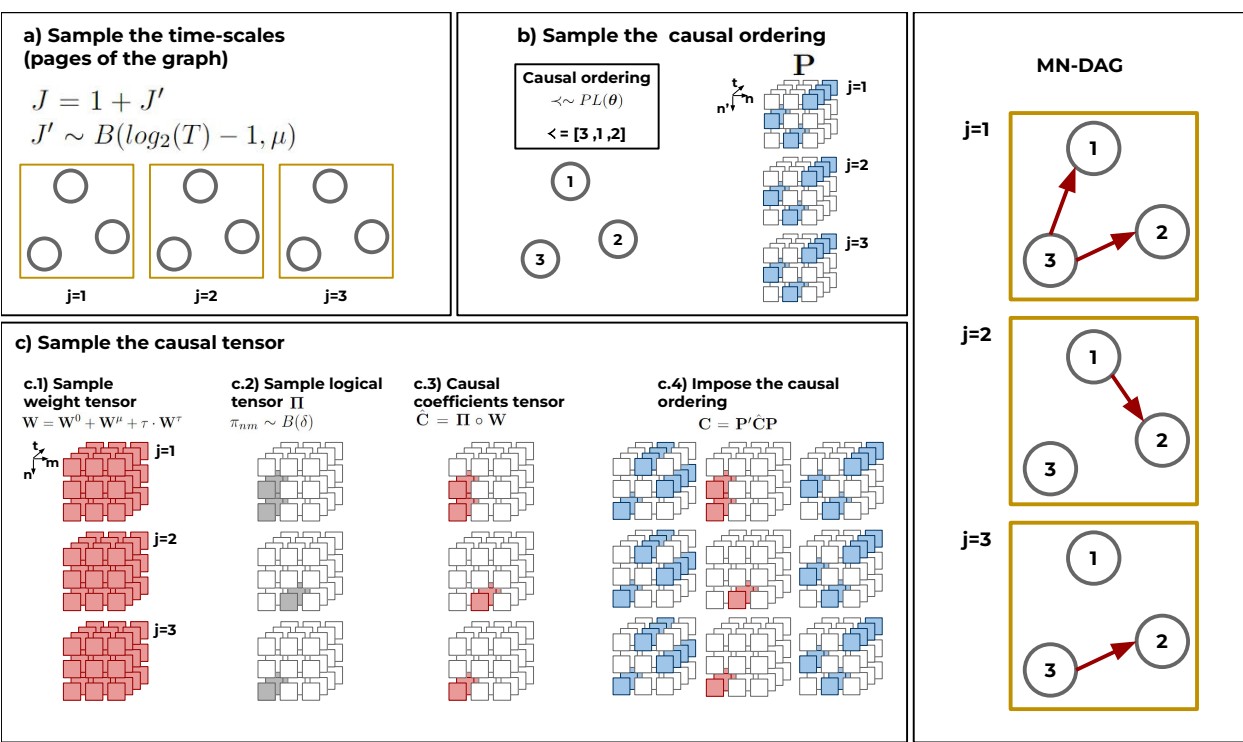

Figure 2: The figure shows the steps necessary to sample a MN-DAG. For the sake of readability, let us consider the case where N=3 and T=4. Here the yellow color refers to time scales; red for non-stationarity. (a) First, we sample the number of pages (time scales) of the MN-DAG. Given $\mu$, the latter is given by $J = 1 + J'$, where $J' \sim B(log_2(T) - 1, \mu)$. In the example, we instantiate three pages ($J = 3$). (b) Second, we sample the causal ordering $\prec \sim PL(\boldsymbol{\theta})$ that is shared by all time scales and entails the permutation tensor $\mathbf{P} \in \{0,1\}^{J \times T \times N \times N}$. Here, $PL(\boldsymbol{\theta})$ indicates the Plackett-Luce distribution, defined by a score vector $\boldsymbol{\theta} \in \mathbb{R}^N$, where $\theta_i \sim U(0, N)$. The indexes are $j$ for time scales, $t$ for time steps, $n$ for the considered nodes and $n'$ for the positions within $\prec$. In the considered example, we have $\prec = [3, 2, 1]'$. Therefore, for each $3 \times 3$ slice of the tensor corresponding to a certain time scale $j$ and time $t$, we have $p_{n'n} = 1$ (blue square) if the node $n$ appears at index $n'$ within $\prec$. (c) Third, we build the tensor of causal coefficients as follows. With regards indexes, $j$, $t$ and $n$ are the same as above, whereas $m$ indicates the parents dimension. Here, we first sample a full tensor of weights $\mathbf{W} \in \mathbb{R}^{J \times T \times N \times N}$ made by three components: (i) a constant term $\mathbf{W}^0$; (ii) $\mathbf{W}^\mu$ that makes the magnitude of causal relationships different across time scales; (iii) $\mathbf{W}^\tau$ that allows causal coefficients to vary over time according to batched $GP(0, \mathbf{K})$. Therefore, within each scale $j$, $W_{nm}$ are smooth functions varying over index $t$. To manage the density of the entailed MN-DAG, we multiply element-wise $\mathbf{W}$ by a logical mask $\boldsymbol{\Pi} \in \{0,1\}^{J \times T \times N \times N}$. The entries of the $\boldsymbol{\Pi}$ are distributed according to a Bernoulli distribution, $\pi_{nm} \sim B(\delta)$. Finally, we obtain the causal tensor $\mathbf{C}$ that entails the MN-DAG on the right by imposing the causal ordering sampled at step (b).

permutation of a vector of integers $\prec' = [1, \ldots, N]$, thus we use the *Plackett-Luce* distribution (PL, Plackett 1975; Luce 1959) to sample it. $PL$ represents a distribution over permutations, defined by a vector of scores $\boldsymbol{\theta} \in \mathbb{R}^N$, which allows sampling permutations $\mathbf{b} \in \mathcal{S}_N$ in $\mathcal{O}(N \log N)$, where $\mathbf{b}$ is a vector of $N$ integers and $\mathcal{S}_N$ is the support of permutations of $N$ elements. Thus, given $\boldsymbol{\theta}$, the probability of a permutation $\mathbf{b}$ is

$$p(b \mid \theta) = \prod_{i=1}^{k} \frac{e^{\theta_{b_i}}}{\sum_{u=i}^{k} e^{\theta_{b_u}}} .$$

A sample $\mathbf{b}$ from $PL$ distribution can be thought as a sequence of samples from categorical distributions: first $b_1$ comes from the categorical distribution with logits $\boldsymbol{\theta}$; $b_2$ from the categorical with logits $\boldsymbol{\theta} - \{\theta_{b_1}\}$; and so on. The mode of the $PL$ is the descending order permutation of scores $\mathbf{b}^0 = \theta_{b_1^0} \geq \theta_{b_2^0} \geq \ldots \geq \theta_{b_N^0}$. The sampling procedure from a $PL$ relies upon the fact that an order of a vector $\mathbf{z} \in \mathbb{R}^N \sim Gum(\boldsymbol{\theta}, 1)$ is

distributed as $PL(\boldsymbol{\theta})$, where $Gum(\boldsymbol{\theta}, 1)$ is a Gumbel distribution with location parameter $\boldsymbol{\theta}$ and scale equal to one (Gadetsky et al., 2020). Therefore, we can sample **b** as follows:

$$z_i = \theta_i - \log(-\log(v_i)), \qquad v_i \sim U(0, 1)$$
$$H(\mathbf{z}) = \mathrm{argsort}(\mathbf{z}).$$

We sample the causal ordering $\prec \sim PL(\boldsymbol{\theta})$ by using the procedure above, where we choose a uniform prior for the $PL$ score vector, i.e., $\theta_i \sim U(0, N)$. The causal ordering $\prec$ entails a permutation matrix $\widehat{\mathbf{P}} \in \{0, 1\}^{N \times N}$ such that $p_{n'n} = 1$ iff the variable $X_n$ occurs at position $n'$ within $\prec$; 0 otherwise. Finally, we derive a permutation tensor $\mathbf{P} \in \mathbb{R}^{J \times T \times N \times N}$ by simply tiling $\widehat{\mathbf{P}}$ along both multiscale and time dimensions.

**Sample the causal tensor.** Given $J$ and $\tau$, we build a tensor of weights $\mathbf{W} \in \mathbb{R}^{J \times T \times N \times N}$ made by three building blocks. First, we sample $\widehat{\mathbf{W}}^0 \in \mathbb{R}^{N \times N}$, whose entries are normally distributed, $w_{nm}^0 \sim N(0, 1)$. Starting from $\widehat{\mathbf{W}}^0$, we derive the first component $\mathbf{W}^0 \in \mathbb{R}^{J \times T \times N \times N}$ by simply expanding $\widehat{\mathbf{W}}^0$ along both multiscale and time dimensions. Then, this component can be thought as a constant term shared by all temporal resolutions and time steps.

Second, we sample $\widehat{\mathbf{W}}^\mu \in \mathbb{R}^{J \times 1 \times N \times N}$, whose entries are distributed according to a Gaussian $N(0, \mu)$. By expanding $\widehat{\mathbf{W}}^\mu$ along the time dimension, we obtain the second component $\mathbf{W}^\mu \in \mathbb{R}^{J \times T \times N \times N}$. This component makes the magnitude of causal relationships different across scales and is stationary along $t$.

Third, we sample $\mathbf{W}^\tau \in \mathbb{R}^{J \times T \times N \times N}$ where each tube along the time dimension follows a multivariate Gaussian distribution $MN(\mathbf{0}, \mathbf{K})$. Here, the covariance matrix $\mathbf{K} = \mathbf{K}(t, t')$ represents a (combination of) valid kernel(s) for Gaussian processes (GP, Bishop & Nasrabadi 2006), where the lengthscale is $\lambda = 1/\tau$. This component imposes the causal coefficient to evolve smoothly over time, according to $\tau$. Indeed, as $\tau \to 0$, the lengthscale of the kernel increases and consequently $\mathbf{W}^\tau$ varies more slowly in time. Finally, the tensor of weights is

$$\mathbf{W} = \mathbf{W}^0 + \mathbf{W}^\mu + \tau \cdot \mathbf{W}^\tau. \tag{1}$$

Now, to manage sparsity and ensure the acyclicity of causal connections, we generate a suitable logical mask $\widehat{\mathbf{\Pi}} \in \{0, 1\}^{J \times 1 \times N \times N}$. Indeed, we use this mask to obtain a tensor of causal relations from $\mathbf{W}$, made of strictly lower triangular slices. The slices $\widehat{\mathbf{\Pi}}_{nm}$ are strictly lower triangular and the entries are distributed according to a Bernoulli distribution, $\pi_{nm} \sim B(\delta)$. Then, we obtain the tensor of causal relations as $\widehat{\mathbf{C}} = \mathbf{\Pi} \circ \mathbf{W}$, whose slices $\widehat{\mathbf{C}}_{nm}$ are nilpotent[1]. Here $\mathbf{\Pi} \in \{0, 1\}^{J \times T \times N \times N}$ is obtained by expanding $\widehat{\mathbf{\Pi}}$ over time and $\circ$ represents the Hadamard product.

At this point, given $\mathbf{P}$ and $\widehat{\mathbf{C}}$, we compute the causal tensor that entails the latent MN-DAG by means of the product $\mathbf{C} = \mathbf{P}'\widehat{\mathbf{C}}\mathbf{P}$, where $\mathbf{P}'$ is obtained by transposing the two rightmost dimensions of $\mathbf{P}$.

## 5 Generate Data from the MN-DAG

Having sampled an MN-DAG, we wish to use it to generate $N$ zero-mean processes of length $T$, whose behaviour is determined by the evolution over time of a latent MN-DAG. Here, we build upon the SEM (Appendix A) and the MLSW (Appendix C) theoretical frameworks. Mathematically, we model the multivariate time series as

$$\mathbf{X}_T[t] = \sum_{j=1}^{J} \sum_{k=-\infty}^{+\infty} \mathbf{M}_j[\nu]\mathbf{z}_{j,k}\psi_j[t-k]. \tag{2}$$

In Equation (2), (i) $\{\psi_j[t-k]\}$ is a set of non-decimated wavelets (see Appendix B); (ii) $\{\mathbf{z}_{j,k}\}$ is a set of random vectors $\mathbf{z}_{j,k} \sim N(\mathbf{0}, \mathbb{I}_{N \times N})$; (iii) $\mathbf{M}_j[\nu] = (\mathbb{I} - \mathbf{C}_j[\nu])^{-1}$ is a time-dependent mixing matrix that represents the transfer function matrix, where $\nu = k/T$ is the rescaled time (Dahlhaus, 1997) and

---

[1] A matrix $\mathbf{A}$ is said nilpotent if it is square and $\mathbf{A}^{\bar{n}} = 0$ for all integers $\bar{n} \geq \bar{N}$, where $\bar{N}$ is known as the index of $\mathbf{A}$.

$\mathbf{C}_j[\nu] \in \mathbb{R}^{N \times N}$ is the matrix of causal coefficients at time $\nu$ and scale $j$ described in Section 4. In our model, the local variance and cross-covariance between the processes at a certain time $\nu$ and scale $j$, i.e., the *local wavelet spectral matrix* (LWSM) $\mathbf{S}_j[\nu]$ is determined by the MN-DAG structure. The same holds for the inverse of the LWSM, i.e., $\mathbf{O}_j[\nu] = \mathbf{S}_j[\nu]^{-1}$, that provides information concerning the local linear dependence between two time series after having removed the linear effects of the rest of the time series. Thus, we can think of it as the equivalent of the precision matrix in the time-frequency domain. Then, the matrix $\mathbf{O}_j[\nu]$ relates to partial correlations between time-series, a concept that has been extensively studied in spectral theory (Dahlhaus 2000, and references therein). Indeed, rescaling leads to the partial coherence $\mathbf{R}_j[\nu] = -\mathbf{D}_j[\nu]\mathbf{O}_j[\nu]\mathbf{D}_j[\nu]$, with $\mathbf{D}_j[\nu]$ being a diagonal matrix with entries $[\mathbf{O}_j[\nu]]_{nn}^{-1/2}$, $\forall n \in [N]$. Given its definition, the partial coherence between two time series provides a measure of local linear dependence as well and is bounded in $[-1, 1]$. The results below link the spectral properties of $\mathbf{X}_T$ to the underlying multiscale causal structure.

**Lemma 5.1.** *The transfer function matrix is a permuted lower triangular matrix,* $\mathbf{M}_j[\nu] = \mathbf{P}'(\mathbb{I}-\widetilde{\mathbf{C}}_j[\nu])^{-1}\mathbf{P}$, *where* $\mathbf{P} \in \mathbb{R}^{N \times N}$ *is a permutation matrix such that* $p_{n'n} = 1$ *iff the node* $X_n$ *occurs at position* $n'$ *within the causal ordering* $\prec$, *and* $\widetilde{\mathbf{C}}_j[\nu]$ *is a strictly lower triangular matrix of causal coefficients.*

*Proof.* See Appendix E.

Equipped with Lemma 5.1, we obtain the following.

**Lemma 5.2.** *The local wavelet spectral matrix and its inverse are given by* $\mathbf{S}_j[\nu] = \mathbf{P}'(\mathbb{I} - \widetilde{\mathbf{C}}_j[\nu])^{-1}(\mathbb{I} - \widetilde{\mathbf{C}}_j[\nu])^{-1'}\mathbf{P}$ *and* $\mathbf{O}_j[\nu] = \mathbf{P}'(\mathbb{I} - \widetilde{\mathbf{C}}_j[\nu])'(\mathbb{I} - \widetilde{\mathbf{C}}_j[\nu])\mathbf{P}$.

*Proof.* See Appendix E.

Lemma 5.2 provides us with the expressions of $\mathbf{S}_j[\nu]$ and $\mathbf{O}_j[\nu]$ in terms of the features of the MN-DAG, that are the causal ordering entailing the permutation matrix $\mathbf{P}$ and the matrix of causal coefficients. From a computational perspective, the expression $\mathbf{O}_j[\nu]$ is more appealing than that of $\mathbf{S}_j[\nu]$ since it does not involve any matrix inversion. Therefore in Section 6 we leverage $\mathbf{O}_j[\nu]$, given its convenient form and the information that it provides.

It is interesting to understand how the causal ordering and the order in which we observe the dimensions of $\mathbf{X}_T$ impact the spectral properties of the process, i.e., the information contained in $\mathbf{S}_j[\nu]$ and $\mathbf{O}_j[\nu]$. Proposition 5.3 shows that any permutation of the causal matrix leaves the spectral properties unchanged. In addition, it proves that any re-ordering of the dimensions of $\mathbf{X}_T$ results in a representation of the form in Equation (2) with the same spectral properties.

**Proposition 5.3.** *The spectral properties of the process* $\mathbf{X}_T$ *are independent of both the causal ordering and the order in which the process dimensions are observed.*

*Proof.* See Appendix E.

# 6 Two-Step Inference

We expose a Bayesian method for the estimation of MN-DAGs, termed MN-CASTLE. It is implemented by using Pyro (Bingham et al., 2019) a probabilistic programming language built on Python and Py-Torch (Paszke et al., 2019). A probabilistic model is a stochastic function that generates data $\mathbf{x}$ according to latent random variables $\mathbf{z}$ and parameters $\boldsymbol{\beta}^*$, having as joint density function

$$p_{\boldsymbol{\beta}^*}(\mathbf{x}, \mathbf{z}) = p_{\boldsymbol{\beta}^*}(\mathbf{x} \mid \mathbf{z})p_{\boldsymbol{\beta}^*}(\mathbf{z}),$$

where $p_{\boldsymbol{\beta}^*}(\mathbf{z})$ and $p_{\boldsymbol{\beta}^*}(\mathbf{x} \mid \mathbf{z})$ are the prior and the likelihood, respectively. The goal is to learn the parameters of the model $\boldsymbol{\beta}^*$ from data. As detailed in Appendix D, SVI offers a scheme to learn $\boldsymbol{\beta}^*$ by approximating the usually intractable posterior distribution $p_{\boldsymbol{\beta}^*}(\mathbf{z} \mid \mathbf{x})$ by means of a tractable family of variational distributions $q_{\boldsymbol{\phi}}(\mathbf{z})$, called guides, parameterized by the variational parameters $\boldsymbol{\phi}$.

Our task is as follows. We are given a dataset $\mathcal{X} = \{\mathbf{X}_T[t]\}_{t=1}^T$, $\mathbf{X}_T[t] = [X_T^1[t], \ldots, X_T^N[t]]'$ and an estimate of the inverse LWSM $\widehat{\mathbf{O}}_j$ at different time scales $j$. As an example, the inverse smoothed bias-corrected raw wavelet periodogram is a suitable non-parametric estimator (Park et al., 2014). Then, according to the probabilistic generative model in Section 5, we want to learn the following parameters given previous inputs by means of SVI: (i) the vector of scores $\boldsymbol{\theta}$ of the Plackett-Luce distribution used to model the latent global causal ordering $\prec$; (ii) the mean and kernel parameters of the latent batched GPs used to model the entries of the hidden causal coefficients tensor $\mathbf{C}$, i.e., $C_j^{(n,m)} \sim GP(\bar{C}_j^{(n,m)}, \mathbf{K}(t, t'))$. Here, we assume that functional form of the kernel $\mathbf{K}(t, t')$ is shared by all causal the coefficients. In addition, by learning the kernel parameters, we obtain an estimate $\hat{\tau}$ of $\tau$ since we assume $\tau = 1/\lambda$ as in Section 4.

In light of Lemma 5.2 and Proposition 5.3, we set the inference of the causal ordering apart from that of the causal coefficients. Indeed, our inference procedure is as follows:

- *Step 1*: We estimate the parameter vector $\hat{\theta}$ of the PL distribution which determines the causal ordering by conditioning on the dataset $\mathcal{X}$;
- *Step 2*: We estimate the parameters of the kernels associated with the causal coefficients by conditioning on the estimate $\widehat{\mathbf{O}}_j$, while imposing the causal ordering from Step 1.

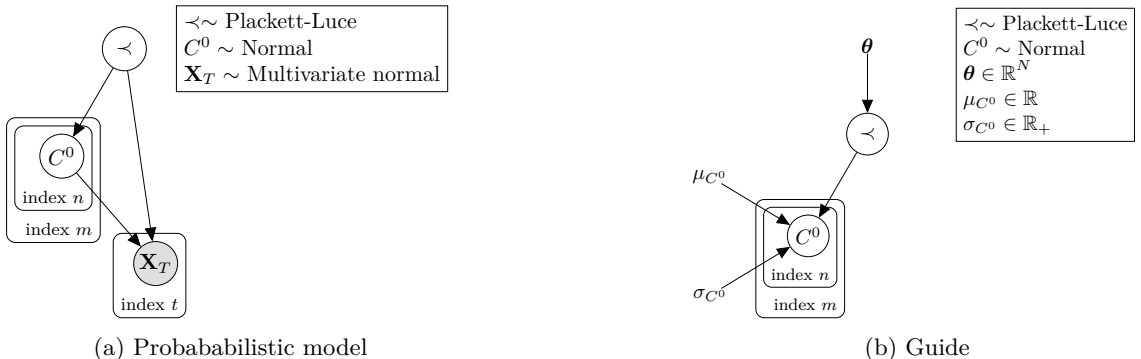

(a) Probabililistic model          (b) Guide

Figure 3: Graphical models associated with a the probabilistic model and b the parameterized variational distribution for learning the causal ordering, along with variational parameters and their constraints.

**Model and guide for causal ordering inference.** Figures 3a and 3b provide a pictorial representation of the probabilistic model and the guide used in Step 1. In particular, we resort to graphical models to illustrate the corresponding joint distributions. Here, random variables are represented as circular nodes, where a blank node represents a latent variable while a grey one is associated with an observed variable. Deterministic variables are represented as rhomboid nodes, while variational parameters are printed outside of the nodes. Edges indicate dependence among variables and rectangles (plates) indicate conditionally independent dimensions, i.e., independent copies. In addition, Figure 3 provides the distributions (along with the constraints of parameters) of random variables and variational parameters.

Figure 3 shows that model and guide share the same latent variables. Indeed, since the guide is used to approximate the true posterior, it needs to provide a valid probability density over all hidden variables. More in detail, we have two latent variables: (i) the causal ordering, which is global since it does not depend on any other variable and is modeled within the guide as a $PL(\boldsymbol{\theta})$; (ii) a stationary single-scale causal structure $\mathbf{C}^0 \in \mathbb{R}^{N \times N}$, where each entry $C_{nm}^0$ is independent of the others and is modeled in the guide with a Gaussian. Since we assume the causal ordering (i) shared by all time scales and (ii) stationary; we infer it from observed data $\mathbf{X}_T$, without any additional information concerning the variance decomposition and its evolution over different temporal resolutions (provided by a given estimate of $\widehat{\mathbf{S}}_j$). For this reason, to learn $\boldsymbol{\theta}$, we resort to the SEM formulation given in Equation (4), where we set $\mathbf{C}^0 = \mathbf{P}'\widehat{\mathbf{C}}^0\mathbf{P}$ (see Section 4). As a consequence, the causal tensor $\mathbf{C}^0$ in Figure 3 depends on $\prec$. According to the probabilistic model in Figure 3a, at each time-step we observe the vector $\mathbf{X}_T[t]$ by using a multivariate normal likelihood,

precisely $MN(\mathbf{0}, \mathbf{MM}')$. Indeed, with constant causal coefficients and normally distributed noises, we have: (i) $\mathbb{E}[\mathbf{X}_T[t]] = \mathbf{M} \cdot \mathbb{E}[\mathbf{Z}[t]] = \mathbf{M} \cdot \mathbf{0} = \mathbf{0}$; (ii) $\text{Var}[\mathbf{X}_T[t]] = \text{Var}[\mathbf{MZ}] = \mathbf{M}\mathbb{I}\mathbf{M}' = \mathbf{MM}'$.

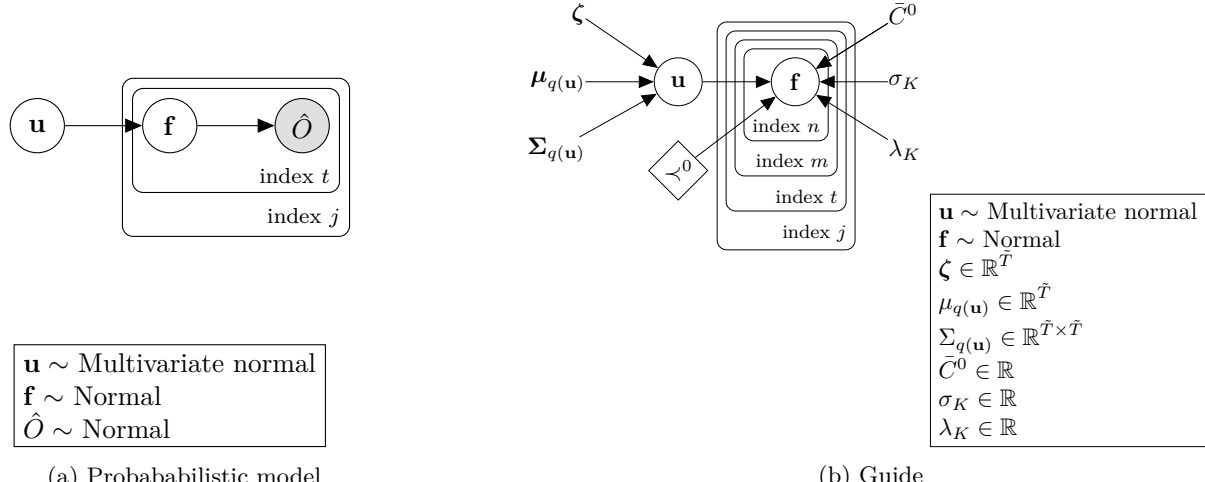

(a) Probabilistic model             (b) Guide

Figure 4: Graphical models associated with a the probabilistic model and b the parameterized variational distribution for learning batched GPs, along with variational parameters and their constraints.

**Model and guide for batched GPs inference.** Figures 4a and 4b depict the probabilistic model and the guide used in Step 2. Here we exploit the result from Step 1. We compute the mode of the PL distribution $\prec^0 = \text{argsort}(-\hat{\boldsymbol{\theta}})$, and then build the permutation matrix $\mathbf{P}$ as described in Section 4. Since known, we represent $\prec^0$ with a rhomboid node.

To model each latent causal coefficient at a certain temporal resolution level $j$ as a smoothly varying function $\mathbf{f} \sim GP(\bar{C}_j^{(n,m)}, \mathbf{K}(t, t'))$, we exploit a variational formulation of Gaussian processes (Hensman et al., 2015). Accordingly, we consider a set of inducing points $\boldsymbol{\zeta} = \{\zeta_{\tilde{t}}\}_{\tilde{t}=1}^{\tilde{T}}$ optimised over the training set, where $\tilde{T} \le T$, and latent inducing function variables $\mathbf{u}$ (a subset of $\mathbf{f}$) over these inducing points. Thus, the method relies upon the introduction of a joint variational distribution $q(\mathbf{f}, \mathbf{u})$, such that it factorises as $p(\mathbf{f} \mid \mathbf{u})q(\mathbf{u})$ (Titsias, 2009). This allows to avoid the computation of $\mathbf{K}_{\mathbf{ff}}^{-1}$ within the inference procedure. Here, to approximate the true GP prior $p(\mathbf{u})$ over the inducing points, we choose $q(\mathbf{u})$ to be a Cholesky variational distribution, i.e., a multivariate normal with positive definite covariance matrix $MN(\boldsymbol{\mu}_{q(u)}, \boldsymbol{\Sigma}_{q(u)})$. This variational approach allows to reduce the computational burden of GP estimation while avoiding overfitting at the same time (Bauer et al., 2016). In detail, in our work the usage of $\tilde{T}$ inducing points lowers the computational cost of each GP from $\mathcal{O}(T^3)$ to $\mathcal{O}(\tilde{T}^3)$ (Hensman et al., 2015). As a consequence, in both Figures 4a and 4b we have two latent variables: $\mathbf{u}$ associated with inducing functions and $\mathbf{f}$ associated with GP prior values. Here, we use batched GP to model causal coefficients, consequently they are independent both within and among time scales (rectangles in Figure 4). Since the joint distributions within the model and the guide $q(\mathbf{f}, \mathbf{u})$ factorise as;

$$p(\mathbf{f}, \mathbf{u}) = p(\mathbf{f} \mid \mathbf{u})p(\mathbf{u}); \qquad q(\mathbf{f}, \mathbf{u}) = p(\mathbf{f} \mid \mathbf{u})q(\mathbf{u}),$$

we also draw edges from $\mathbf{u}$ to $\mathbf{f}$. The variational parameters, along with their constraints, are shown in Figure 4b.

Now, let us consider a lower triangular matrix of ones $\mathbf{L}$, having size $N \times N$. Furthermore, define $\bar{\mathbf{R}}_j \in \mathbb{R}^{N \times N}$ such that the entries are $[\bar{\mathbf{R}}_j]_{nn'} = \|[\mathbf{R}_j]_{nn'}\|_2 = (\sum_{k'=1}^{T} [\mathbf{R}_j]_{nn'}[k'/T]^2)^{1/2}$. Hence, we mask the distribution of the hidden functions with $\mathbf{B}_j = (\mathbf{P}'\mathbf{LP}) \circ \bar{\mathbf{R}}_j$ to take into account and update only the relationships conforming to $\prec^0$ and associated with nonzero partial correlation, where $\circ$ represents the Hadamard product. In practice, since it is difficult to exactly estimate zero partial correlation, before constructing the mask it is possible to hard-threshold the values of $\bar{\mathbf{R}}_j$ at a certain value $\rho \in [0, 1]$. In our experiments we set $\rho = 0.05$. At this point, we observe the estimated $\hat{\mathbf{O}}_j$ by using a Gaussian likelihood $N((\mathbb{I} - \mathbf{C}_j)'(\mathbb{I} - \mathbf{C}_j), \sigma)$, where the scale $\sigma \in \mathbb{R}_+$ is fixed (here we use 0.05). In particular, the mean value of the latter Gaussian is set

in accordance with Lemma 5.2. To implement these probabilistic model and guide, we combine Pyro and GPyTorch (Gardner et al., 2018), an efficient Python library for GP inference built on PyTorch.

**Inference.** We optimize the variational parameters above by using SVI, and adopting as optimizer Adam (Kingma & Ba, 2014) along with learning rate decay and gradient clipping (Goodfellow et al., 2016). These tricks are useful to avoid bouncing around local optima when you are close to them and to prevent the gradient from becoming too large. In Step 1, we optimize w.r.t. $\boldsymbol{\theta}$ to approximate the likelihood of $\prec$ given $\mathbf{X}_T$. Unfortunately, the latent variable $\prec$ is non-reparameterizable. Therefore, we use the REINFORCE estimator (Williams, 1992), which is suitable for getting Monte-Carlo estimates of a certain cost function $f_\phi(\mathbf{z})$. According to REINFORCE, we have

$$\nabla_\phi \mathbb{E}_{q_\phi(\mathbf{z})}[f_\phi(\mathbf{z})] = \mathbb{E}_{q_\phi(\mathbf{z})}[(\nabla_\phi \log_\phi(\mathbf{z})) f_\phi(\mathbf{z}) + \nabla_\phi f_\phi(\mathbf{z})]. \tag{3}$$

Although unbiased, this estimator is known to have high variance. A way for reducing this variance is by means of control variate strategies, i.e., by adding a function within the expectation operator in Equation (3) that depends on the chosen values for $\mathbf{z}$ but is constant w.r.t. $\boldsymbol{\phi}$. So, the additional term does not affect the mean of the gradient estimator. Here, we resort to a data dependent baseline (Mnih & Gregor, 2014). The rationale behind the usage of baselines, is to reduce the variance by tracking the mean value of $f_\phi(\mathbf{z})$. Thus, we add a running average of $f_\phi(\mathbf{z})$, namely $\overline{f_\phi(\mathbf{z})}$, for predicting the value of $f_\phi(\mathbf{z})$ at each step. On the contrary, in Step 2 we exploit the reparameterization trick (see Appendix D), to the benefit of learning. Finally, we return the learned variational parameters once the maximum number of iterations is reached.

## 7 Results

In this section we present the empirical assessment of our proposal. We first dive in the statistical analysis of the time series generated by the proposed model in Section 7.1. Then, Section 7.2 presents results regarding the inference of MN-DAGs from synthetic data.

### 7.1 Probabilistic Model over MN-DAGs

We start by illustrating the output of the proposed probabilistic generative model by means of an example. We consider $N = 3$ nodes (time series), $T = 512$ time steps, multiscale level $\mu = 0.5$, non-stationarity level $\tau = 0.5$, and density of causal interactions $\delta = 0.5$.

First, Figure 5 displays the underlying MN-DAG, sampled as detailed in Section 4, along with the evolution over time of causal relationships. We obtain an MN-DAG composed of three pages, corresponding to temporal resolutions $2^j$, $j = \{1, 2, 3\}$. The sampled causal ordering is $\prec = [1, 3, 2]$, and all causal relations, here locally periodic functions with increasing variations, are compliant with $\prec$. Indeed, we can only observe directed edges from time series $n$ to $m$, where $n \prec m$.

Now, given the sampled MN-DAG, we generate data according to Equation (2), where we use non-decimated Haar wavelet (Nason et al., 2000) as oscillatory function $\psi_j[t-k]$. Figure 6 depicts the generated time series along with descriptive statistics.

On the first row, we have the behaviour over time of synthetic data. Here, we resort to the augmented Dickey-Fuller test (ADF, Dickey & Fuller 1979) to assess stationarity. According to the test, the null hypothesis $H_0$ indicates that the process has a unit root (i.e., is non-stationary). The resulting $p$-values prove that the generated processes are (weakly) stationary (we reject $H_0$). Indeed, they have zero mean, while their dispersion looks different. On the one hand, the variance of $X_1$ (which occur at first position in $\prec$) is stationary, on the other hand those of $X_2$ and $X_3$ vary over time. Furthermore, $X_2$, that has incoming causal edges at all temporal resolutions, displays the largest swings.

On the second row we provide the histograms of the data, where we employ a Jarque-Bera test (JB, Jarque & Bera 1987) to assess normality. In particular, the null hypothesis $H_0$ is that the process is normally distributed. The resulting $p$-values suggest that $X_1$ is normally distributed, while we reject $H_0$ for both $X_2$ and $X_3$. Indeed, the associated distributions are leptokurtic, with $X_3$ having a more pronounced negative fat tail.

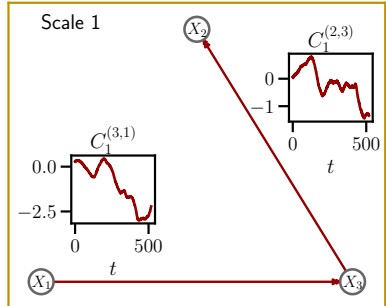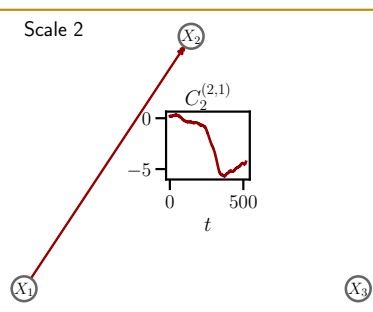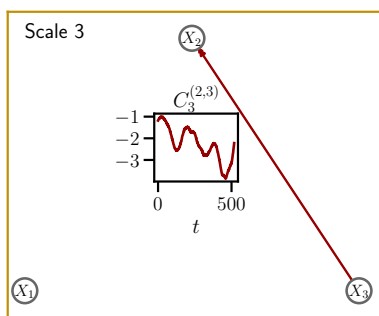

Figure 5: The figure depicts the latent MN-DAG sampled by using the proposed probabilistic generative model, where we set the number of nodes $N = 3$, number of time steps $T = 512$, multiscale level $\mu = 0.5$, non-stationarity level $\tau = 0.5$, and density parameter $\delta = 0.5$. The resulting MN-DAG has (i) 3 nodes; (ii) $J = 3$ pages (yellow rectangles), (iii) non-stationary causal interactions (red directed arrows, values shown as time series in the insets) that follow a Gaussian process with kernel $K = K_{\text{Periodic}} + K_{\text{Linear}} \times K_{\text{Matern}^{3/2}}$; (iv) global causal ordering $\prec = [1, 3, 2]$. Within each scale, we also plot the evolution of causal relations over time. Kernel variances are $\sigma_{\text{Linear}} = \sigma_{\text{Periodic}} = \sigma_{\text{Matern}^{3/2}} = 1$; the lengthscales $\lambda_{\text{Periodic}} = \lambda_{\text{Matern}^{3/2}} = 1/\tau$, and the period $\rho_{\text{Periodic}} = 1/\tau$. Given the kernel shape, the causal coefficients are locally periodic functions with increasing variation.

Looking at the autocorrelation (with lag $l \in [1, 40]$) plotted on the third row, we see that all the generated time series show serial correlation, statistically significant at 95% level (light blue bands). This result is in accordance with the multiscale nature of the time series. In addition, the autocorrelation is driven by the local wavelet spectral matrix $\mathbf{S}_j$ (see Appendix C), that in our model is determined by the causal structure.

Finally, the autocorrelation of absolute values of the processes prove that large swings in $X_2$ and $X_3$, either negative or positive, tend to be followed by other large swings. This effect is also known as volatility clustering, a key-feature of financial time series (Mandelbrot, 1967; Ding & Granger, 1996). Here, large movements in the series are driven by the increase of causal coefficients modulus, shown in Figure 5.

## 7.2 Causal Structure Learning from Multiscale Data with Time-dependent Variance

We next report a comparison between our method, MN-CASTLE, the algorithm for multiscale causal structure learning introduced by D'Acunto et al. (2022), MSCASTLE, and state-of-the-art algorithms for learning Equation (4). For this comparison we use baselines belonging to different families, and synthetic data generated by the proposed probabilistic generative model. The goal is to assess the gain, in terms of performance, as we deviate from the single-scale stationary case, i.e., $\tau = \mu = 0$, which is the closest to Equation (4). Additionally, we report results concerning the inferred causal ordering.

**Settings.** We run our experiments according to four main different configurations. First, to evaluate the methods as we move within the $(\tau, \mu)$-quadrant, we generate the data by setting $N = 5$ and $T = 100$, while the entries of the $PL$ score vector are drawn from a uniform distribution $\theta_i \sim U(0, N)$. We test three values each for the multiscale and non-stationarity parameters, thus giving raise to configurations of none, medium, and high values for each parameter. For each possible combination $(\tau, \mu) \in \{0.0, 0.5, 0.9\} \times \{0.0, 0.5, 0.9\}$, we generate 20 datasets that contain $N$ time series each of length $T$. With regards the causal structure density, we use $\delta = 0.5$.

Second, to measure the sensitivity of the performances w.r.t. network density, we set $(N, T, \tau, \mu)$ equal to $(5, 100, 0.5, 0.5)$ and let $\delta$ varies in $\{0.25, 0.5, 0.75\}$. For each possible combination, we generate 20 datasets.

Third, to measure the sensitivity of the performances w.r.t. network size, we set $(T, \tau, \mu, \delta)$ equal to $(100, 0.5, 0.5, 0.25)$ and let $N$ varies in $\{5, 10, 15, 20\}$. Thus, in this experimental context we go from a configuration in which the number of observations $T$ is greater than the number of relationships possible

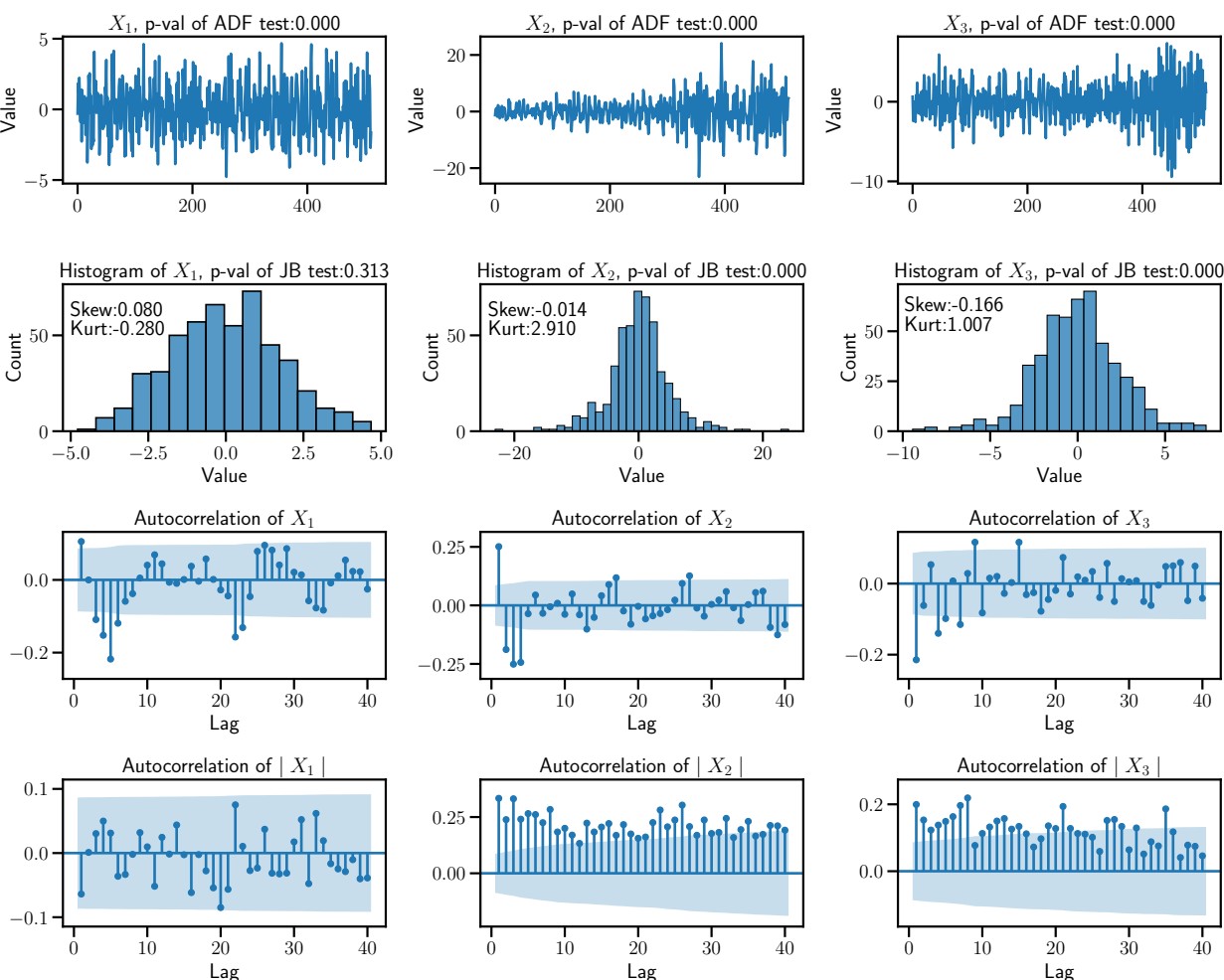

Figure 6: The figure shows the generated time series, along with descriptive statistics, where each process is associated with a different column. (i) Starting from the top, we have the synthetic data obeying to the underlying MN-DAG, where we provide the $p$-values of an ADF test. (ii) On the second row, we have the histograms of observed values, along with the $p$-values of a JB test, skewness, and kurtosis. (iii) The third row shows time series autocorrelations (with lag $l = \{1, \ldots, 40\}$). The light blue bands show 95% CIs. (iv) The last row shows the autocorrelations of absolute values of the processes.

in a complete single-scale DAG, i.e., $N \cdot (N-1)/2$, to one in which it is much less. Also in this case, we generate 20 datasets for each combination.

Fourth, we look at the performances of MN-CASTLE under generative model misspecification, i.e., when the assumptions underlying Equation (2) are violated. Here, we consider two kinds of violations. First, we set $(N, T, \tau, \mu)$ equal to $(5, 100, 0.5, 0.5)$ and violate the Gaussianity assumption of the latent noise $\mathbf{z}_{j,k}$. Indeed, we generate 20 datasets for the case in which $\mathbf{z}_{j,k} \sim L(0,1)$ and 20 for that where $\mathbf{z}_{j,k} \sim U(0,1)$, being $L(0,1)$ the Laplace distribution with zero mean and unit scale and $U(0,1)$ the uniform distribution over the unit interval. Second, we run MN-CASTLE on downsampled data, meaning that the frequency associated with the observational task is lower than that at which causal interactions occur. In detail, we generate 20 datasets by setting $(N, T, \tau, \mu)$ equal to $(5, 256, 0.5, 0.5)$. Then, we subsample the observations to $T = 128$ and run MN-CASTLE on these decimated datasets. This way, our model cannot exploit the information related to the scale level $j = 1$, and consequently cannot infer the causal structure at that time resolution. Hence, we test whether the lack of such information affects the performances of MN-CASTLE in retrieving the causal relationships at coarser time resolutions.

Finally, in case $\tau \neq 0$, for the GP we use the radial basis function kernel $K_{\mathrm{RBF}}$ with variance $\sigma_{\mathrm{RBF}} = 0.1$ and lengthscale $\lambda_{\mathrm{RBF}} = 1/\tau$.

**Baselines.** We test MN-CASTLE against the following four baseline models. First we consider MSCAS-TLE, a multiscale causal structure learning model which exploits multiresolution analysis and non-convex continuous optimization to retrieve stationary causal relationships. Next, we have DirectLiNGAM (Shimizu et al., 2011), a method belonging to the family of non-Gaussian models. Algorithms within this class assume that the noise $\mathbf{Z}$ is non-normally distributed. Indeed, in this case the causal structure has shown to be fully identifiable (Shimizu et al., 2006). Then, DirectLiNGAM returns an estimation of both causal ordering and causal coefficients. Second, we have CD-NOD (Huang et al., 2020), which belongs to the family of constraint-based methods. In particular, it has been developed to deal with heterogeneous (no assumptions on data distributions and causal relations) and non-stationary data as well. GOLEM (Ng et al., 2020) lives at the intersection of score-based and gradient-based methods. It solves an unconstrained optimization problem where the objective function is given by a likelihood function (as in score-based methods), penalized by regularization terms for sparsity and acyclicity.

As already mentioned, the concept of multiscale, non-stationary causal graphs is an understudied topic. Since none of the previous baseline models have been developed to infer causal graphs from data obeying to an underlying MN-DAG, our results provide information regarding the robustness of the previous algorithms with respect to the presence of multiple time scales and non-stationarity. In the following experiments, we use the code of MSCASTLE developed by D'Acunto et al. (2022); we exploit the implementations of DirectLiNGAM and GOLEM provided by `gCastle`[2] (Zhang et al., 2021a), whereas we resort to `causallearn`[3] for the implementation of CD-NOD. The configuration for each baseline is provided in Appendix F.

Differently from MN-CASTLE, baseline models are non-probabilistic. While our model provides an approximate predictive posterior distribution over MN-DAGs, baseline models return a point estimate of an acyclic causal structure. In order to compare the algorithms, we retain all causal coefficients identified by MN-CASTLE that are in modulus significantly greater than 0.1 at 99% level.

**Performance in the estimation of the adjacency tensor.** Retrieving the adjacency tensor means identifying the presence of causal relations disregarding their intensity. Appendix G provides an example of the evolving causal relations inferred by MN-CASTLE, while Appendix H gives insights concerning the estimate of the non-stationarity parameter. Figure 7 refers to the first configuration described above, and shows the F1 score (the higher the better) for the considered models. The definition of the considered metrics are given in Appendix I. Appendix J also reports the values for additional evaluation scores and the fraction of undirected edges for MN-CASTLE.

Given a $(\tau, \mu)$ setting, for each model we have 20 values of F1. We use the box plot in order to visualize the inter-quartile range (IQR). In addition, within the box plot we overlay the F1 scores attained for each of

---

[2] https://github.com/huawei-noah/trustworthyAI
[3] https://github.com/cmu-phil/causal-learn

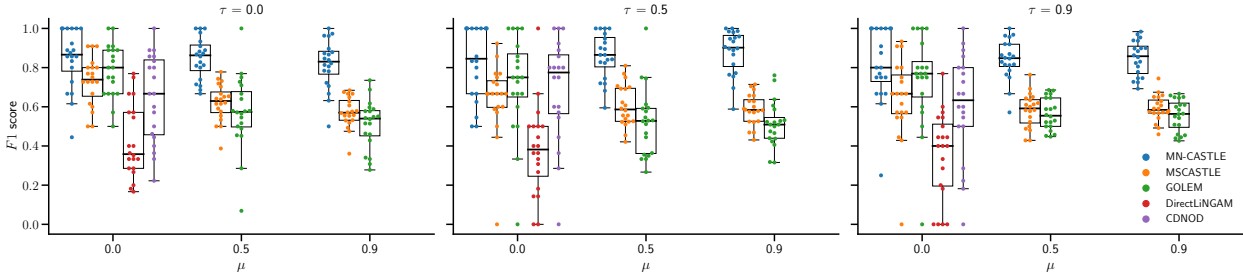

Figure 7: The figure depicts the performances of the considered methods in the retrieval of the adjacency tensor, according to F1 score. Higher F1 indicates better performance. Each model is associated to a different color. For every $(\tau, \mu)$ setting and every model, we represent the values attained over the 20 synthetic datasets through a box plot. Within the latter, we overlay the performances obtained for each dataset (points).

the 20 datasets, plotted as points. For each value of $\tau$, we provide the performances of each algorithm as $\mu$ varies. In case $\mu \neq 0$, we return the performance of GOLEM as well. In particular, because $\prec$ is global, we replicate the causal structure retrieved by the latter for each time scale. This way, we obtain an additional baseline method also for multiscale datasets.

Overall, MN-CASTLE outperforms the baseline models in each case. When $\mu = 0$, the performances of MSCASTLE and GOLEM is very similar to that of our method. In contrast, as $\mu$ increases, we observe that the gap between MN-CASTLE and the other models widens and that, in general, the two multiscale models outperform GOLEM. In addition, MN-CASTLE performance remains stable as $\tau$ increases, and when $\mu$ increases the IQR tightens.

Figure 8a depicts the performances of the models as we vary the density of the underlying MN-DAG (second setting). Since here we use $\mu = 0.5$, we only retain MN-CASTLE, MSCASTLE and GOLEM. MN-CASTLE outperforms the baseline models along $\delta$. In addition, the larger $\delta$, the better the performances of our model.

Figure 8b provides the performances of the methods as we vary the size of the underlying MN-DAG (third setting). Since $\mu = 0.5$, we only compare MN-CASTLE, MSCASTLE and GOLEM also here. MN-CASTLE consistently outperforms the baseline models along $N$. The overall downtrend in the performances of the considered methods stems from the growth of the dimensionality of the problem while the number of observations is kept fixed.

Last but not least, Figure 8c depicts the performances of MN-CASTLE under model misspecification. The results prove that our model behaves as good as the case with no violation in every scenario.

**Performance in the estimation of $\boldsymbol{\theta}$.** Figure 9 provides results concerning the goodness of the inferred vector of scores $\hat{\boldsymbol{\theta}}$ of $PL$ distribution for the first experimental configuration, as measured by normalized discounted cumulative gain (nDCG) at 3, w.r.t. the ground truth causal ordering $\prec$. Appendix I provides insights concerning the computation of this metric, whereas Appendix J shows the results for Kendall-$\tau$ statistics, Spearman's rank correlation and nDCG at 5. To represent the results, we use box plots. For each of the 20 synthetic datasets, generated according to specific a pair $(\tau, \mu)$, we obtain an estimated $\hat{\boldsymbol{\theta}}$. Then, we sample $10^3$ causal orderings $\hat{\prec}$ from $PL(\hat{\boldsymbol{\theta}})$. Now, for each drawn causal ordering, we evaluate the monitored metric w.r.t. $\prec$. As vector of scores for a baseline model, we use $\bar{\boldsymbol{\theta}}$, where $\bar{\theta}_i \sim U(0, N)$, $i = 1, \ldots, N$. Afterwards, we obtain $10^3$ random causal orderings $\bar{\prec}$ by sampling from the $PL$ distribution parameterized by $\bar{\boldsymbol{\theta}}$. As for MN-CASTLE, we evaluate the metric w.r.t. $\prec$. Therefore, for each model, every box plot is built by using $2 \times 10^4$ points. Overall, according to the monitored metric, MN-CASTLE outperforms the baseline model. In addition, the performances do not deteriorate as $\tau$ grows and improve as $\mu$ increases.

We apply the same approach to evaluate the sensitivity of the estimation accuracy for $\boldsymbol{\theta}$ with respect to the density and number of nodes of the underlying MN-DAG and under generative model misspecification. Figure 10 shows the results obtained on the synthetic data generated according to the second, third, and fourth experimental settings, described above.

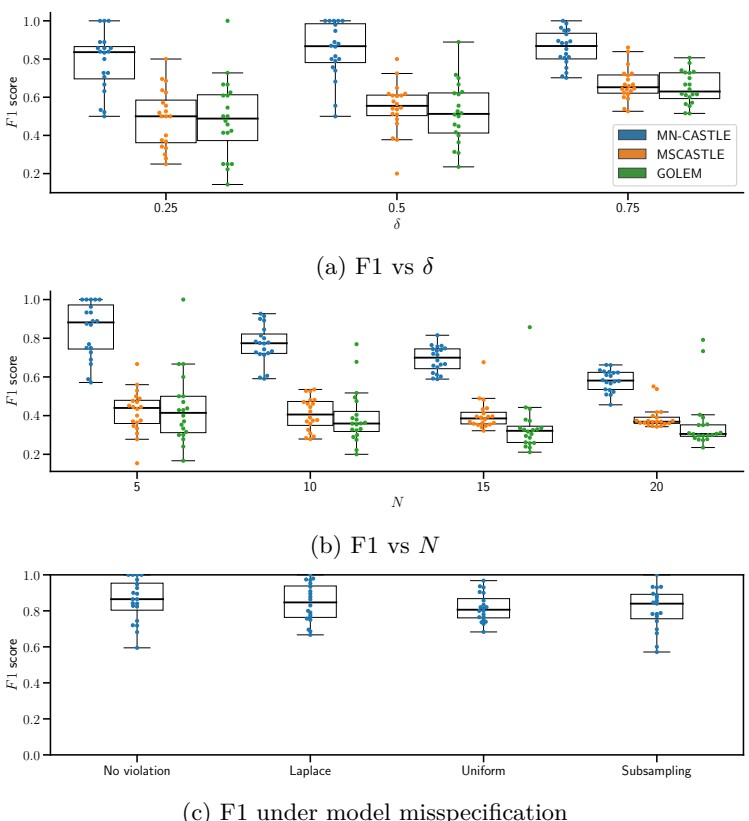

(a) F1 vs $\delta$

(b) F1 vs $N$

(c) F1 under model misspecification

Figure 8: The figure depicts the performances of the considered methods in the retrieval of the adjacency tensor, according to F1 score, along (a) $\delta$ and (b) $N$, and (c) under generative model misspecification. In the latter setting, to ease the comparison we also report the performances for the case without violation. Higher F1 indicates better performance. Each model is associated to a different color. We represent the values attained over the 20 synthetic datasets through box plots, where we overlay the performances obtained for each dataset (points).

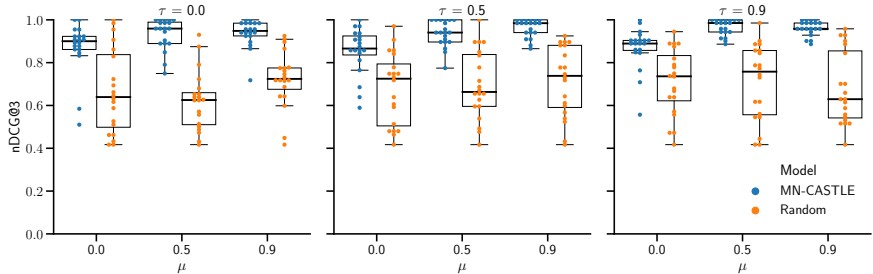

Figure 9: The figure depicts box plots along with quartiles reference lines (dashed lines) for normalized discounted cumulative gain (nDCG) at 3. MN-CASTLE is given in blue while a random baseline model in orange. For every dataset generated according to a given $(\tau, \mu)$ setting (i) we sample $1 \times 10^3$ causal orderings $\hat{\prec} \sim PL(\hat{\boldsymbol{\theta}})$, where $\hat{\boldsymbol{\theta}}$ is the estimated vector of scores; (ii) we draw $1 \times 10^3$ random causal orderings $\bar{\prec} \sim PL(\bar{\boldsymbol{\theta}})$, where $\bar{\theta}_i \sim U(0, N)$, $i = 1, \ldots, N$. Afterwards, we evaluate nDCG@3 by using the sampled causal orderings and $\prec$ for both models. Thus, each box plot is made by $2 \times 10^4$ points.

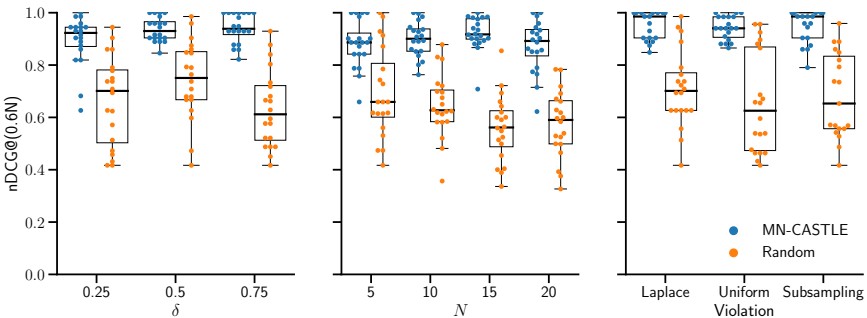

Figure 10: The figure depicts box plots for the normalized discounted cumulative gain (nDCG) at $0.6N$. MN-CASTLE is given in blue while a random baseline model in orange. Box plots on the left refer to the second experimental configuration, i.e., when we vary $\delta$ while keeping fixed the values of the others parameters, as described above. Box plots on the center concern the third experimental setting, where we study the sensitivity of the estimation accuracy w.r.t. the network size $N$. Box plots on the right relate the fourth experimental setting, where we study MN-CASTLE under generative model misspecification. Notice that in the left and right plots $N = 5$.

Overall, the accuracy of MN-CASTLE in retrieving $\boldsymbol{\theta}$ grows along with $\delta$: the IQR of the monitored metric reach higher values. In addition, the performance of MN-CASTLE in recovering the causal ordering is high and shows no dependence on $N$. Notice that the nDCG score depends on the value of $N$. Thus, the comparison is meaningful because it considers the same fraction of nodes for each combination. Finally, the performance of MN-CASTLE does not degrade under model violations.

# 8 Analysis of Natural Gas Prices in the US Market

In this section, we examine the key drivers of natural gas prices in the US market during the period spanning from January 1, 2018, to December 31, 2022. Our analysis considers several variables, including the price of natural gas (NG), crude oil (CO), deviations in gas storage (SD), rig counts targeting gas (RC), deviations from seasonal average values of gas consumption for cooling (CDD) and heating (HDD) environments, the crack spread between heating oil and crude oil (CS), and the economic uncertainty index (UI, Baker et al., 2016). We collected the data on a weekly basis, and we analyze time scales ranging from 1 to 5, corresponding to resolutions ranging from 2-4 (scale 1) to 32-64 (scale 5) weeks. Please refer to Appendix K for further details on data sources and the pre-processing of the time series data.

The algorithms used for this dataset are GOLEM, MSCASTLE, and MN-CASTLE. GOLEM identifies stationary instantaneous interactions between variables, meaning relationships that occur at a frequency higher than weekly and remain constant over time. It does not establish any causal relationship with NG, as the only interaction detected is SD→HDD.

MSCASTLE is capable of detecting causal interactions between the time series on the 5 considered time scales. However, these interactions are assumed to be stationary over time. Figure 11 shows the multiscale causal networks learned by MSCASTLE. The edges associated with a positive causal coefficient are represented in green, while those with a negative coefficient are represented in pink. The thickness of the edge is directly proportional to the absolute value of the causal coefficient. Unlike GOLEM, the multiscale analysis allows MSCASTLE to detect interactions that occur at longer time resolutions. The method suggests that NG causes CO on both scale 1 and 2, with positive causal coefficient. While it is known from the literature that the price of CO drives the price of NG due to the substitution processes of NG with petroleum products, the NG→CO relationship represents a novel element difficult to justify in light of the interchangeable relationships between CO and NG. Like GOLEM, MSCASTLE detects a negative causal interaction from SD to HDD, which occurs on the first, third and fourth scales. On the contrary, on the second scale the relationship is reversed. Overall, the interactions between SD and HDD are the larger in magnitude.

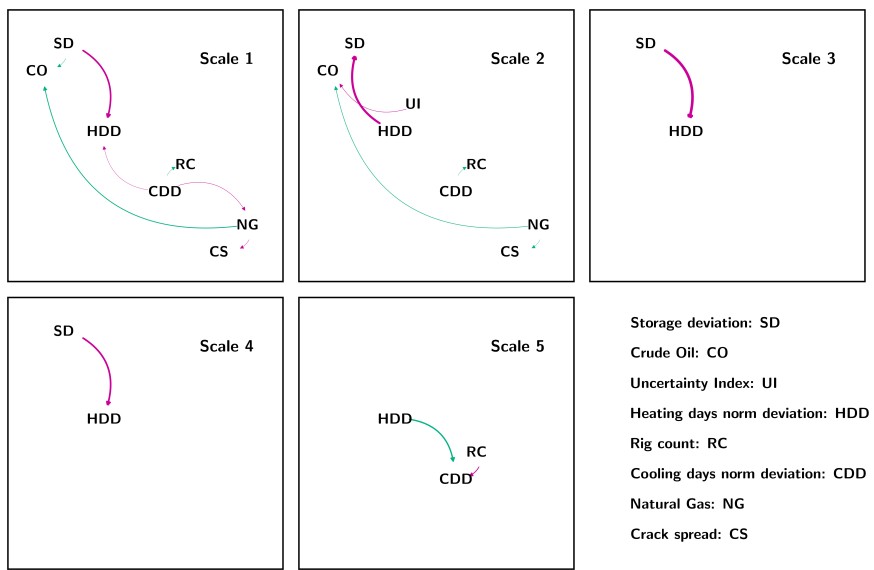

Figure 11: The figure depicts the multiscale DAG retrieved by MSCASTLE

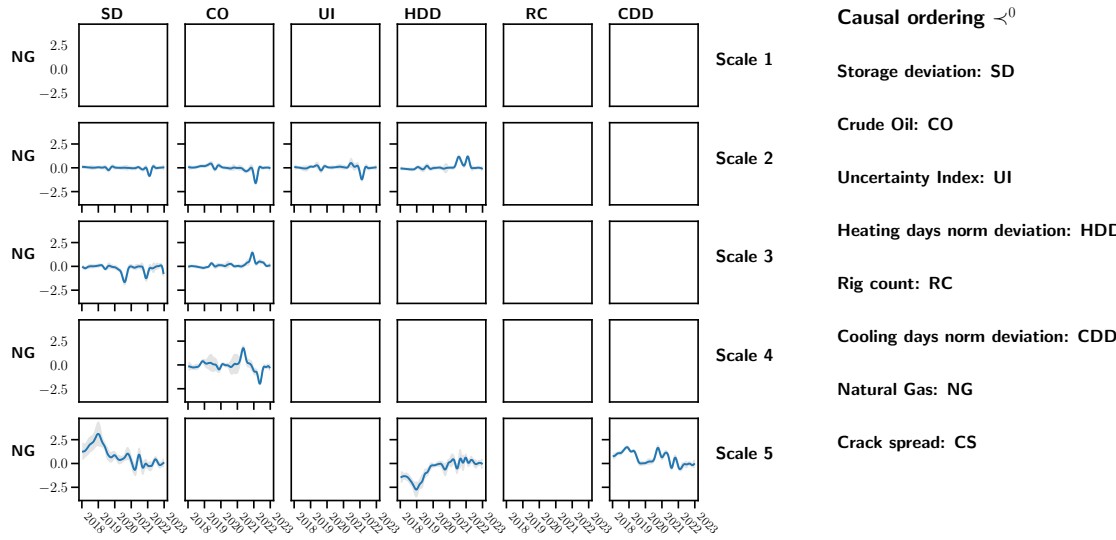

Figure 12: The figure depicts the relationships inferred by MN-CASTLE which involve NG at the considered scales, and the estimated causal ordering $\prec^0$.

MN-CASTLE can detect causal relationships between variables at different temporal resolutions, allowing for more nuanced insights into how the variables interact over time. The behavior of the causal coefficients learned by MN-CASTLE is illustrated in Figure 12. For readability, here we only report the coefficients involving NG, which is the focus of our analysis. The full MN-DAG is given in Appendix L. Overall, the information provided by our method is richer and more detailed than the baselines. This is due to the ability of our method to track the temporal evolution of causal connections and detect relationships that are activated in a limited period of time and that may change sign. As a result, MN-CASTLE suggests denser causal structures, thus providing a more comprehensive understanding of how the variables are related.

Interestingly, many of the causal relationships inferred by MN-CASTLE intensify during two specific periods: the early months of 2020 and 2022. These time windows correspond to significant events that put pressure on the energy market: the outbreak of COVID-19 and the Russian invasion of Ukraine. The fact that MN-CASTLE can detect these changes in causal relationships highlights its ability to capture the dynamic nature of the market and the impact of exogenous events.

Concerning the causal ordering, in Figure 12 we observe that SD, CO, and UI occupy the first three positions, whereas NG and CS occupy the last ones. Our method is the only one able to detect that the price of NG is causally influenced by seasonal factors, economic uncertainty, oil price, and deviations in gas reserves. These relationships occur at larger temporal resolutions of 4 weeks, indicating that short-term fluctuations have a limited impact on NG prices. Our findings are consistent with the literature (Brown & Yucel, 2008; Nick & Thoenes, 2014; Ji et al., 2018), which highlights the relevance of economic and environmental factors in shaping energy markets. In line with the estimated causal ordering, MN-CASTLE does not propose the NG→CO interaction returned by MSCASTLE. Furthermore, the absence of interactions at lower scales is in line with the fact that GOLEM does not detect any connection regarding gas and underlines the superiority of multiscale methods. Regarding the impact of SD on NG, we see that at scales 2 and 3, an increase/decrease in SD causes a decrease/increase in NG prices, thus suggesting that supply shocks in the mid term can affect NG prices. On the contrary, on scale 5, we observe a positive relationship around the years 2018-2019. This relationship can be explained by the fact that the demand for NG increased in that period (also due to exports) thus causing an increase in the price of NG, while gas reserves increased due to the record production of NG in the US.

MN-CASTLE detects the causal interaction of CO→NG at different time scales and highlights changes in sign over the analyzed period. Specifically, scales 3 and 4 show that in 2021, the increase in CO prices causes a corresponding increase in NG prices. However, in 2022, scales 2 and 4 indicate a negative relationship, possibly due to geopolitical tensions and speculative phenomena.

In addition, we observe that the prices of NG tend to be higher during periods with greater deviations in HDD and CDD, which represent unusual weather events.

## 9 Conclusions and Future Research Directions

This paper deals with multiscale non-stationary causal analysis, filling a gap in the literature. Indeed, the bulk of previous work assumes that the only relevant temporal resolution for causal relations is the frequency of observed data. We drop such assumption. In addition, we also allow the causal relations to vary over time. Since in general there is no prior knowledge about the relevant time scales of causal interactions nor about their temporal dependencies, the proposed framework of MN-DAGs represents an important step in such direction.

**Generative model.** We propose a probabilistic model to generate time series data obeying to an underlying MN-DAG, in accordance with the specified values for multiscale and non-stationary features, $\mu$ and $\tau$, respectively. Our model leverages the well established mathematical theory of multivariate locally stationary wavelet processes and linear structural equation model. The causal ordering is modeled by means of Plackett-Luce distribution while the causal interactions evolve over time according to the specified kernel of a Gaussian process. Statistical analysis of generated data proves the exposed model to be able to reproduce well-known features of time series. Therefore, it represents a suitable framework for testing the robustness of causal

structure learning methodologies on datasets generated from different points of the $(\tau, \mu)$-quadrant shown in Figure 1.

We stress the importance of providing both researchers and practitioners with synthetic data generators capable of replicating phenomena characterizing data from different application domains.

Future work should aim to overcome some limitations related to the framework adopted to manage different time resolutions and the modeling of the causal tensor. In particular, multivariate locally stationary wavelet processes formulation relies upon wavelets, that are known to suffer from limited joint time-frequency resolution (Heisenberg uncertainty principle). Indeed, wavelets divide the frequency space into non-overlapping bands, i.e., octave bands. Furthermore, since the auto/cross-correlation structure of generated data depends on both the power spectrum decomposition across temporal scales and the auto-correlation wavelet, the usage of diverse wavelet families might lead to different results. Then, the usage of alternative methods to wavelet transform might improve the proposed generative model. With regards the causal tensor, structural breaks such as sudden deletion/addition of causal edges might be added within Equation (1).

From a theoretical point of view, an interesting research direction is to study the assumptions that make the model described in Eq. (2) identifiable. Even though some class of linear structural equation models have been proved identifiable under different types of restrictions (Shimizu et al., 2006; Peters & Bühlmann, 2014; Loh & Bühlmann, 2014; Park & Kim, 2020), the case of MN-DAG needs to be carefully investigated. Indeed, the presence of the non-decimated wavelet transform; the unobservability of the contributions to the process coming from each time resolution; the linearity of the model in the frequency domain are some of the points that distinguish the MN-DAG case from those currently studied.

**Bayesian causal structure learning method.** In addition, we expose a Bayesian method for learning MN-DAGs from time series data, termed MN-CASTLE. The latter relies upon observed time series data and an estimate for the inverse power spectrum at each scale level. We implement the latter by using a two-step approach. In the first step we optimize w.r.t. the Plackett-Luce vector of scores $\boldsymbol{\theta}$, by using the values of time series at time $t$. Then, we keep the causal ordering fixed to the mode of the Plackett-Luce distribution, i.e., $\prec^0$, and we estimate the rest of variational parameters related to the causal coefficient tensor by exploiting the provided estimation for the inverse power spectrum.

Our findings show that MN-CASTLE compares favorably to baseline models in the retrieval of the adjacency tensor of the causal graph. We test the models on synthetic datasets generated according to different $(\tau, \mu)$ configurations, from the single-scale stationary to highly multiscale non-stationary case. We observe that the performance of MN-CASTLE, depicted in Figure 7, is not sensitive to the value of non-stationarity parameter. On the contrary, the growth of the multiscale parameter is associated with an improvement in the quality of the results returned by our method since the IQR tightens. The improvement in performances along $\mu$ is also shown in Figure 9, that concerns the goodness of the estimated vector of scores for Plackett-Luce distribution. On one hand, we think that when non-stationarity and multiscale parameters are different from zero, MN-CASTLE might benefit of greater differences among time series distributions. On the other hand, we believe that the large variance shown (especially in the single scale case) is an effect due to the low cardinality of the edge set. In fact, even though the monitored metric is normalized, on average in the single scale case we only have five causal links. So, a single error weighs more. We emphasize that MN-CASTLE, being a fully Bayesian approach, by definition takes into account uncertainty. Consequently, we might sample MN-DAGs from the approximate posterior distribution in accordance with the confidence of the model.

Furthermore, we study the behaviour of our model w.r.t. the density $\delta$ of the underlying MN-DAG. We observe that the performance of MN-CASTLE improves as $\delta$ increases and that our method outperforms the other models in all cases. This improvement is also manifest in the value of the metric used to evaluate the estimated causal ordering. We also provide supplementary results on additional synthetic data to test the capabilities of the monitored methods when the MN-DAG size $N$ increases, keeping the other parameters fixed. Also in this settings, MN-CASTLE outperforms the baselines. In addition, the ability of MN-CASTLE in estimating causal ordering does not deteriorate as N increases. Last but not least, MN-CASTLE keeps performing well under model misspecification, thus highlighting the robustness of our method.

In our case study, we have applied MN-CASTLE to analyze the drivers of natural gas in the US market and have compared the results of our method with those of GOLEM and MSCASTLE. While we cannot determine the ground-truth, we have found that MN-CASTLE provides richer information than the baselines, as it can track the evolution of causal relationships across different scales over time. Our study has revealed that causal relationships have strengthened during the outbreak of COVID-19 and at the beginning of the Russian invasion of Ukraine. Additionally, by accounting for non-stationarity, MN-CASTLE has detected more relationships than the baselines. Furthermore, our method is the only one capable of detecting the causal impact of seasonal factors, economic uncertainty, oil prices, and deviations in gas storage on natural gas prices, which are crucial drivers in the Economics literature.

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

## Appendix A    Linear Structural Equation Models and DAGs

Mathematically, a DAG is formulated as a SEM. Given a dataset $\mathcal{X} := (x_1, \ldots, x_N)$ of $N$ random variables, a SEM is a collection of $N$ structural assignments

$$x_i := f_i(\mathcal{P}_i, z_i), \quad i = 1, \ldots, N,$$

where $\mathcal{P}_i$ represents the set of direct causes (parents) of node $x_i$, $z_i$ is a noise variable satisfying $z_i \perp z_j$ if $j \neq i$, and $f_i(\cdot)$ is a generic functional form. In this paper, we focus on linear functional forms, therefore, by exploiting matrix form, the equation above becomes:

$$\mathbf{X} = \mathbf{CX} + \mathbf{Z},$$

where $\mathbf{C} \in \mathbb{R}^{N \times N}$ is the matrix of causal coefficients satisfying (i) $c_{ii} = 0 \quad \forall i \in \{1, \ldots, N\}$; (ii) $c_{ij} \neq 0 \iff x_j \in \mathcal{P}_i$; (iii) $diag(\mathbf{C}^n) = \mathbf{0}, \quad \forall n \in \mathbb{N}$ (acyclicity property). Since $\mathbb{I} - \mathbf{C}$ is an invertible matrix (see Lemma Appendix E.1), we can rewrite the latter equation as

$$\mathbf{X} = \mathbf{MZ}, \tag{4}$$

with $\mathbf{M} = (\mathbb{I} - \mathbf{C})^{-1}$ being a mixing matrix. According to Equation (4), observed data is a mixing of independent latent noises. Here, causal relations are stationary, instantaneous and are supposed to occur at the frequency of observed data.

## Appendix B    Locally Stationary Wavelet Process

In the univariate case, *locally stationary wavelet* process (LSW, Nason et al. 2000) is a suitable modeling framework to represent a non-stationary process $\mathbf{x}_T$ of length $T = 2^J, \mathrm{J} \in \mathbb{N}$, by means of a triangular multiscale representation

$$x_T[t] = \sum_{j=1}^{J} \sum_{k=-\infty}^{+\infty} v_j[k/T] z_{j,k} \psi_j[t-k]. \tag{5}$$

The building blocks of Equation (5) are: (i) the random amplitude $v_j[k/T] z_{j,k}$ composed by a time-varying amplitude $v_j[k/T]$ and a normal noise variable $z_{j,k}$ such that $cov(z_{j,k}, z_{j',k'}) = \widetilde{\delta}_{j,j'} \widetilde{\delta}_{k,k'}$, where $\widetilde{\delta}_{j,j'}$ represents the Kronecker delta; (ii) discrete, real valued and compactly-supported oscillatory functions $\psi_j[t-k]$, namely *non-decimated wavelets*. At each time only some values contribute to $x_T[t]$, and the time-dependence is managed by the index $k$. Local stationarity means that the statistical properties of the process vary slowly over time. This feature is essential in order to make learning possible (Nason et al., 2000). Within the LSW framework, local stationarity is formalized by means of a smoothness assumption concerning the time-varying amplitudes $v_j[k/T]$ (Fryzlewicz et al., 2003). Indeed, the latter quantity provides a measure of the time-dependent contribution to the variance at a certain time scale level $j \leq J$, namely the *evolutionary wavelet spectrum* (EWS), defined as $S_j[\nu] = |v_j[\nu]|^2$, with $\nu = k/T$ being the rescaled time (Dahlhaus, 1997). For a stationary process, EWS is constant $\forall j \leq J$. As an example, consider the $MA(1) = 1/\sqrt{2}(\epsilon[t] - \epsilon[t-1])$. We obtain it by setting in Equation (5) the following values for the previous components: (i) $z_{j,k} = \epsilon[t]$; (ii) $S_j = 1$ if $j = 1$ and zero otherwise; $\psi_1 = [1/\sqrt{2}, -1/\sqrt{2}]$ as the Haar wavelet. Because $S_1$ is constant and different from zero only for $j = 1$, we obtain a stationary amplitude $w_1[\nu] = 1$ only for the first scale level. Then, it follows that

$$\begin{aligned}
x_T[t] &= \sum_{j=1}^{J} \sum_{k=-\infty}^{+\infty} v_j[k/T] z_{j,k} \psi_j[t-k] \\
&= \sum_{k=-\infty}^{+\infty} 1 \cdot \epsilon_{1,k} \psi_1[t-k] \\
&= \frac{1}{\sqrt{2}} (\epsilon[t] - \epsilon[t-1]).
\end{aligned}$$

## Appendix C    Multivariate Locally Stationary Wavelet Process

The MLSW framework generalizes *locally stationary wavelet* process (LSW, Nason et al. 2000; see Appendix B) to model $N$ zero-mean processes $\mathbf{X}_T[t] = [X_1[t], \ldots, X_N[t]]'$, each of length $T$, as follows:

$$\mathbf{X}_T[t] = \sum_{j=1}^{J} \sum_{k=-\infty}^{+\infty} \mathbf{V}_j[k/T]\mathbf{z}_{j,k}\psi_j[t-k]. \tag{6}$$

In Equation (6), (i) $\{\psi_j[t-k]\}$ is a set of non-decimated wavelets; (ii) $\{\mathbf{z}_{j,k}\}$ is a set of random vectors $\mathbf{z}_{j,k} \sim N(\mathbf{0}, \mathbb{I}_{N \times N})$; (iii) $\mathbf{V}_j[k/T] \in \mathbb{R}^{N \times N}$ is the transfer function matrix, assumed to be lower triangular and with entries being Lipschitz continuous functions associated with Lipschitz constants $L_j^{(n,m)}$, $n \in \{1, \ldots, N\}$, $m \in \{1, \ldots, N\}$, such that $\sum_j L_j^{(n,m)} < \infty$. Local stationarity means that the statistical properties of the process vary slowly over time. This feature is essential in order to make learning possible (Nason et al., 2000), and within MLSW coincides with the Lipschitzianity assumption above. Here, the transfer function matrix $\mathbf{V}_j[\nu]$, with $\nu = k/T$ being the rescaled time (Dahlhaus, 1997), provides a measure of the local variance and cross-covariance between the processes at a certain time $\nu$ and scale $j$, i.e., the *local wavelet spectral matrix* (LWSM) $\mathbf{S}_j[\nu] = \mathbf{V}_j[\nu]\mathbf{V}_j'[\nu]$.

By construction, LWSM is symmetric and positive at each time $\nu$ and scale $j$. Within LWSM, diagonal elements $S_{nn}^j[\nu]$ represents the spectra of of the processes, whereas $S_{nm}^j[\nu]$ provides the cross-spectra between them. In addition, the local auto and cross-covariance functions, namely $c_{nn}(\nu, l)$ and $c_{nm}(\nu, l)$ (with $l$ being a certain lag), admit a formulation in terms of the LWSM (see Park et al. 2014 for further details).

## Appendix D    Overview of SVI

*Stochastic variational inference* (Hoffman et al., 2013; Kingma & Welling, 2013) is an algorithm that combines *variational inference* (VI, Blei et al. 2017) and *stochastic optimization* (Spall, 2005). SVI approximates the posterior distribution of complex probabilistic models that involves hidden variables, and can handle large datasets. Consider a dataset $\mathcal{X} = \{\mathbf{x}^{(i)}\}_{i=1}^T$ of $T$ i.i.d. samples of either a continuous or discrete variable $\mathbf{x}$. Suppose that $\mathcal{X}$ is generated according to a latent continuous random variable $\mathbf{z}$, The latter is governed by a vector of parameters $\boldsymbol{\beta}^*$ endowed with a prior distribution $p(\boldsymbol{\beta}^*)$, i.e., $\mathbf{z}^{(i)} \sim p_{\boldsymbol{\beta}^*}(\mathbf{z})$. Thus, we have data are generated according to a conditional distribution, i.e., $\mathbf{x}^{(i)} \sim p_{\boldsymbol{\beta}^*}(\mathbf{x} \mid \mathbf{z})$. Both the prior $p_{\boldsymbol{\beta}^*}(\mathbf{z})$ and the conditional distribution $p_{\boldsymbol{\beta}^*}(\mathbf{x} \mid \mathbf{z})$ belong to parametric families of distributions $p_{\boldsymbol{\beta}}(\mathbf{z})$ and $p_{\boldsymbol{\beta}}(\mathbf{x} \mid \mathbf{z})$ whose PDFs are differentiable w.r.t. $\boldsymbol{\beta}$ and $\mathbf{z}$. Our goal is to compute the likelihood of the hidden variable given the observations, i.e., the posterior

$$p_{\boldsymbol{\beta}}(\mathbf{z} \mid \mathbf{x}) = \frac{p_{\boldsymbol{\beta}}(\mathbf{x}, \mathbf{z})}{\int p_{\boldsymbol{\beta}}(\mathbf{x}, \mathbf{z}) \, d\mathbf{z}}. \tag{7}$$

Since the denominator of Equation (7), also known as evidence, is usually intractable to compute, a well-known solution is to approximate the target posterior. Within approximate posterior inference methodologies, VI casts learning as an optimization problem. More in details, VI involves the introduction of a family of variational distributions $q_{\boldsymbol{\phi}}(\mathbf{z} \mid \mathbf{x})$, parameterized by a variational parameters $\boldsymbol{\phi}$. Then, VI optimizes those parameters to find $q_{\boldsymbol{\phi}^*}(\mathbf{z} \mid \mathbf{x})$, i.e., the member of the variational distributions family that is closest to the posterior distribution. Here closeness is measured according to Kullback-Leibler divergence (KL).

The objective of SVI is the *evidence lower bound* ($ELBO$), that is equal to the negative KL divergence up to a term that does not depend on $q$

$$\begin{aligned} ELBO &= \mathbb{E}_{q_{\boldsymbol{\phi}}(\mathbf{z}|\mathbf{x})}[\log p_{\boldsymbol{\beta}}(\mathbf{x}, \mathbf{z}) - \log q_{\boldsymbol{\phi}}(\mathbf{z} \mid \mathbf{x})] \\ &= -D_{KL}(q_{\boldsymbol{\phi}}(\mathbf{z} \mid \mathbf{x}^{(i)}) \| p_{\boldsymbol{\beta}}(\mathbf{z} \mid \mathbf{x}^{(i)})) + \log p_{\boldsymbol{\beta}}(\mathbf{x}^{(i)}) \\ &= -D_{KL}(q_{\boldsymbol{\phi}}(\mathbf{z} \mid \mathbf{x}^{(i)}) \| p_{\boldsymbol{\beta}}(\mathbf{z})) + \mathbb{E}_{q_{\boldsymbol{\phi}}(\mathbf{z}|\mathbf{x}^{(i)})}[\log p_{\boldsymbol{\beta}}(\mathbf{x}^{(i)} \mid \mathbf{z})]. \end{aligned} \tag{8}$$

Since KL is a non-negative measure of closeness between distributions, then $\log p_{\boldsymbol{\beta}}(\mathbf{x}) \geq ELBO$ for all $\boldsymbol{\beta}$ and $\boldsymbol{\phi}$. Therefore, the maximization of the $ELBO$ is equivalent to the minimization of the distance between

$q_\phi(\mathbf{z})$ and $p_\beta(\mathbf{x} \mid \mathbf{z})$. Observations $\mathbf{x}^{(i)}$ are conditionally independent given the latent, thus the log likelihood term in Equation (8) can be written as

$$\sum_{i=1}^{T} \log p(\mathbf{x}^{(i)} \mid \mathbf{z}) \approx \frac{T}{T'} \sum_{i \in \mathcal{I}_{T'}} \log p(\mathbf{x}^{(i)} \mid \mathbf{z}),$$

where $\mathcal{I}_{T'}$ is a set of indexes of size $T' \leq T$. One way to subsample indexes is, for example, to randomly select $T'$ data points among the observations Thus, in case of large datasets, we can run SVI while exploiting mini-batch optimization.

In order to compute the gradient of the *ELBO* w.r.t. $\phi$, SVI relies upon the reparameterization trick. The continuous random variable $\mathbf{z}$ can be expressed in terms of a deterministic function $\mathbf{z} = g_\phi(\boldsymbol{\epsilon}, \mathbf{x})$, where $\boldsymbol{\epsilon} \sim q(\boldsymbol{\epsilon})$ is independent of $\mathbf{z}$. This procedure is useful to move all the dependence on $\phi$ inside the expectation operator

$$\mathbb{E}_{q_\phi(\mathbf{z} \mid \mathbf{x}^{(i)})}[f_\phi(\mathbf{z})] = \mathbb{E}_{q(\boldsymbol{\epsilon})}[f_\phi(g_\phi(\boldsymbol{\epsilon}, \mathbf{x}^{(i)}))],$$

where $f_{\phi(\mathbf{z})}$ represents a general cost function. Now, the gradient can be computed as

$$\nabla_\phi \mathbb{E}_{q(\boldsymbol{\epsilon})}[f_\phi(g_\phi(\boldsymbol{\epsilon}, \mathbf{x}^{(i)}))] = \mathbb{E}_{q(\boldsymbol{\epsilon})}[\nabla_\phi f_\phi(g_\phi(\boldsymbol{\epsilon}, \mathbf{x}^{(i)}))]$$
$$\approx \frac{1}{L} \sum_{l=1}^{L} f(g_\phi(\boldsymbol{\epsilon}^{(i,l)}, \mathbf{x}^{(i)})),$$

where $L$ is the number of samples per data point. Then, we obtain an unbiased estimate of the gradient by means of Monte-Carlo estimates of this expectation.

## Appendix E   Proofs

**Lemma Appendix E.1.** *The inverse of* $\mathbb{I} - \mathbf{C}$ *exists and consists of a finite sum of powers of* $\mathbf{C}$.

*Proof.* To prove invertibility of $\mathbb{I} - \mathbf{C}$, $\mathbf{C} \in \mathbb{R}^{N \times N}$, let us rewrite $\mathbf{C} = \mathbf{P}'\widetilde{\mathbf{C}}\mathbf{P}$. Here, $\mathbf{P} \in \mathbb{R}^{N \times N}$ is a permutation matrix entailed by the causal ordering $\prec$, such that $p_{n'n} = 1$ iff the node $X_n$ occurs at position $n'$ within $\prec$, and $\widetilde{\mathbf{C}}$ is a strictly lower triangular matrix, computed by ordering the rows of $\mathbf{C}$ according to $\prec$. Now, for permutation matrices it holds $\mathbf{P}^{-1} = \mathbf{P}'$. In addition, since strictly lower triangular matrices are nilpotent, there exists an integer $\bar{N}$ such that $\widetilde{\mathbf{C}}^{\bar{n}} = 0$, $\forall \bar{n} \geq \bar{N}$. Then it follows that $\mathbf{C}$ is similar to $\widetilde{\mathbf{C}}$ and, consequently, nilpotent too:

$$\begin{aligned}
\mathbf{C}^{\bar{N}} &= (\mathbf{P}'\widetilde{\mathbf{C}}\mathbf{P})^{\bar{N}} \\
&= (\mathbf{P}^{-1}\widetilde{\mathbf{C}}\mathbf{P})^{\bar{N}} \\
&= (\mathbf{P}^{-1}\widetilde{\mathbf{C}}\mathbf{P})(\mathbf{P}^{-1}\widetilde{\mathbf{C}}\mathbf{P}) \ldots (\mathbf{P}^{-1}\widetilde{\mathbf{C}}\mathbf{P}) \\
&= \mathbf{P}^{-1}\widetilde{\mathbf{C}}(\mathbf{P}\mathbf{P}^{-1})\widetilde{\mathbf{C}}(\mathbf{P}\mathbf{P}^{-1}) \ldots (\mathbf{P}\mathbf{P}^{-1})\widetilde{\mathbf{C}}\mathbf{P} \\
&= \mathbf{P}^{-1}\widetilde{\mathbf{C}}^{\bar{N}}\mathbf{P} \\
&= 0.
\end{aligned}$$

At this point, exploiting the geometric series representation (nilpotent matrices have eigenvalues equal to zero and then are convergent), we have that

$$\begin{aligned}
(\mathbb{I} - \mathbf{C})^{-1} &= \sum_{\bar{n}=0}^{\infty} \mathbf{C}^{\bar{n}} \\
&= \sum_{\bar{n}=0}^{\bar{N}-1} \mathbf{C}^{\bar{n}}.
\end{aligned}$$

Therefore, the inverse exists and is given by a finite sum of powers of $\mathbf{C}$. $\qquad\square$

**Lemma 5.1.** *The transfer function matrix is a permuted lower triangular matrix, $\mathbf{M}_j[\nu] = \mathbf{P}'(\mathbb{I}-\widetilde{\mathbf{C}}_j[\nu])^{-1}\mathbf{P}$, where $\mathbf{P} \in \mathbb{R}^{N \times N}$ is a permutation matrix such that $p_{n'n} = 1$ iff the node $X_n$ occurs at position $n'$ within the causal ordering $\prec$, and $\widetilde{\mathbf{C}}_j[\nu]$ is a strictly lower triangular matrix of causal coefficients.*

*Proof.* Starting from the representation of $\mathbf{C}_j[\nu] = \mathbf{P}'\widetilde{\mathbf{C}}_j[\nu]\mathbf{P}$, we have:

$$
\begin{aligned}
\mathbf{M}_j[\nu] &= (\mathbb{I} - \mathbf{P}'\widetilde{\mathbf{C}}_j[\nu]\mathbf{P})^{-1} \\
&= (\mathbf{P}^{-1}\mathbf{P} - \mathbf{P}'\widetilde{\mathbf{C}}_j[\nu]\mathbf{P})^{-1} \\
&= (\mathbf{P}'\mathbf{P} - \mathbf{P}'\widetilde{\mathbf{C}}_j[\nu]\mathbf{P})^{-1} \\
&= (\mathbf{P}'(\mathbb{I} - \widetilde{\mathbf{C}}_j[\nu])\mathbf{P})^{-1} \\
&= \mathbf{P}^{-1}(\mathbb{I} - \widetilde{\mathbf{C}}_j[\nu])^{-1}\mathbf{P}^{-\prime} \\
&= \mathbf{P}'(\mathbb{I} - \widetilde{\mathbf{C}}_j[\nu])^{-1}\mathbf{P};
\end{aligned}
$$

where $(\mathbb{I}-\widetilde{\mathbf{C}}_j[\nu])^{-1}$ admits a representation in terms of the geometric series for Lemma Appendix E.1, which in this case consists in a sum of lower triangular matrices. $\qquad\square$

**Lemma 5.2.** *The local wavelet spectral matrix and its inverse are given by $\mathbf{S}_j[\nu] = \mathbf{P}'(\mathbb{I} - \widetilde{\mathbf{C}}_j[\nu])^{-1}(\mathbb{I} - \widetilde{\mathbf{C}}_j[\nu])^{-1'}\mathbf{P}$ and $\mathbf{O}_j[\nu] = \mathbf{P}'(\mathbb{I} - \widetilde{\mathbf{C}}_j[\nu])'(\mathbb{I} - \widetilde{\mathbf{C}}_j[\nu])\mathbf{P}$.*

*Proof.* For real-valued invertible matrix $\mathbf{A}$ the Gramian $\mathbf{A}\mathbf{A}'$ is semi-positive definite, hence $\mathbf{S}_j[\nu] = \mathbf{M}_j[\nu]\mathbf{M}_j[\nu]'$ analogously to Definition 2 in Park et al. (2014).

Then, by pluggin in the expression of $\mathbf{M}_j[\nu]$ given by Lemma 5.1, we get:

$$
\begin{aligned}
\mathbf{S}_j[\nu] &= (\mathbf{P}'(\mathbb{I} - \widetilde{\mathbf{C}}_j[\nu])^{-1}\mathbf{P})(\mathbf{P}'(\mathbb{I} - \widetilde{\mathbf{C}}_j[\nu])^{-1}\mathbf{P})' \\
&= \mathbf{P}'(\mathbb{I} - \widetilde{\mathbf{C}}_j[\nu])^{-1}(\mathbb{I} - \widetilde{\mathbf{C}}_j[\nu])^{-1'}\mathbf{P};
\end{aligned}
$$

where we exploit the fact properties $\mathbf{P}' = \mathbf{P}^{-1}$. By following the same rationale and remembering that $(\mathbf{B}\mathbf{A})^{-1} = \mathbf{A}^{-1}\mathbf{B}^{-1}$, we obtain $\mathbf{O}_j[\nu] = \mathbf{P}'(\mathbb{I} - \widetilde{\mathbf{C}}_j[\nu])'(\mathbb{I} - \widetilde{\mathbf{C}}_j[\nu])\mathbf{P}$ $\qquad\square$

**Proposition 5.3.** *The spectral properties of the process $\mathbf{X}_T$ are independent of both the causal ordering and the order in which the process dimensions are observed.*

*Proof.* Let us consider a causal matrix $\mathbf{C}_j[\nu]$ and its causally ordered form $\widetilde{\mathbf{C}}_j[\nu]$, such that $\mathbf{C}_j[\nu] = \mathbf{P}'\widetilde{\mathbf{C}}_j[\nu]\mathbf{P}$. Now, let us consider $\widetilde{\mathbf{S}}_j[\nu] = (\mathbb{I} - \widetilde{\mathbf{C}}_j[\nu])^{-1}(\mathbb{I} - \widetilde{\mathbf{C}}_j[\nu])^{-1'}$ and $\widetilde{\mathbf{O}}_j[\nu] = (\mathbb{I} - \widetilde{\mathbf{C}}_j[\nu])'(\mathbb{I} - \widetilde{\mathbf{C}}_j[\nu])$. From Lemma 5.2 it directly follows that $\mathbf{S}_j[\nu] = (\mathbb{I} - \mathbf{C}_j[\nu])^{-1}(\mathbb{I} - \mathbf{C}_j[\nu])^{-1'} = \mathbf{P}'\widetilde{\mathbf{S}}_j[\nu]\mathbf{P}$ and $\mathbf{O}_j[\nu] = (\mathbb{I} - \mathbf{C}_j[\nu])'(\mathbb{I} - \mathbf{C}_j[\nu]) = \mathbf{P}'\widetilde{\mathbf{O}}_j[\nu]\mathbf{P}$. This proves independence from the causal ordering.

Now, let us consider a process $\mathbf{X}_T$ admitting a representation of the form of Equation (2). Consider also a permuted version of $\mathbf{X}_T$, i.e., $\widehat{\mathbf{X}}_T = \mathbf{Q}\mathbf{X}_T$ with $\mathbf{Q}$ being an arbitrary permutation matrix of size $N \times N$. By multiplying both sides of Equation (2) by $\mathbf{Q}$, we get that $\widehat{\mathbf{M}}_j[\nu] = \mathbf{Q}\mathbf{M}_j[\nu]$. Hence, we have that $\widehat{\mathbf{S}}_j[\nu] = \mathbf{Q}\mathbf{S}_j[\nu]\mathbf{Q}'$ and $\widehat{\mathbf{O}}_j[\nu] = \mathbf{Q}\mathbf{O}_j[\nu]\mathbf{Q}'$, where $\mathbf{S}_j[\nu] = \mathbf{M}_j[\nu]\mathbf{M}_j[\nu]'$ and $\mathbf{O}_j[\nu] = \mathbf{S}_j[\nu]^{-1}$. This proves independence from the ordering of the dimensions of the process. $\qquad\square$

## Appendix F  Models Configuration

Below we report the models hyper-parameters used during the test phase:

- MN-CASTLE: fraction of inducing points equal to 64%; $K = K_{\mathrm{RBF}}$; in case $\tau = 0$ (the estimated $\hat{O}_j$ is constant) we use as prior for $\lambda_K$ a normal $N(1. \times 10^3, 1. \times 10^{-3})$; $\rho = 0.05$; number of iterations iter$= 6. \times 10^2$ with 10 particles;
- MSCASTLE: $\ell_1-$ penalty parameter $\lambda = 1. \times 10^{-1}$; pruning threshold $\gamma = 5. \times 10^{-2}$; Daubechies wavelet with filter length equal to 2; maximum value for dagness function $h_{\mathrm{tol}} = 1. \times 10^{-8}$;
- GOLEM: pruning threshold $\gamma = 5. \times 10^{-2}$; number of iterations iter$= 1. \times 10^4$;
- DirectLiNGAM: pruning threshold $\gamma = 5. \times 10^{-2}$;
- CDNOD: independence test = Fisher's Z; significance level $\alpha = 95\%$.

## Appendix G  Evolution over Time of the Estimated Causal Relations

We apply MN-CASTLE over a synthetic dataset constituted by $N = 5$ time series of length $T = 100$ each. To generate the data, we use the exposed probabilistic model over MN-DAGs, where we set $\tau = \mu = \delta = 0.5$ and we use as $K = K_{\mathrm{RBF}}$. In this case, we obtain $J = 3$ scale levels. The configuration of MN-CASTLE is the same as Appendix F.

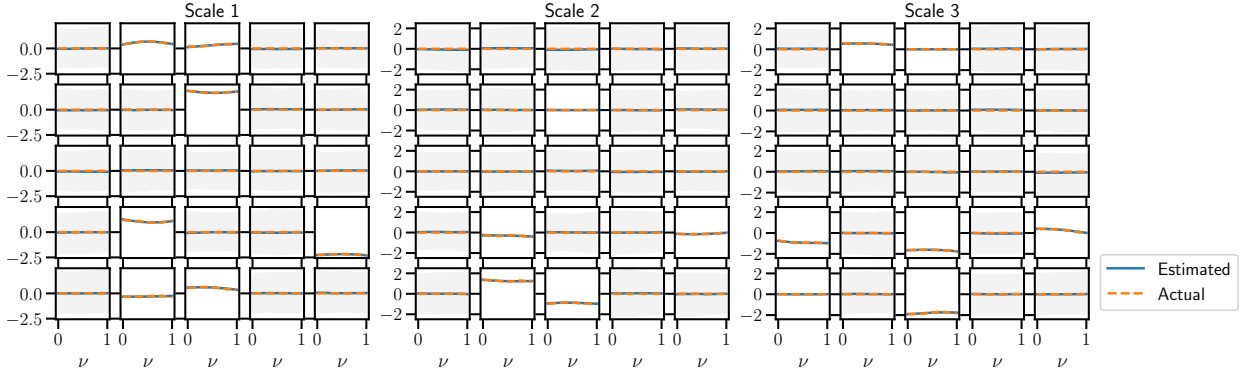

Figure 13: The figure depicts the evolution over time ($\nu = t/T$) of estimated causal coefficients (blue) vs the ground truth latent coefficients (dashed orange), for the three temporal resolutions. Shaded bands refer to 99% confidence level.

Figure 13 depicts the estimated causal relations and their evolution over time. MN-CASTLE correctly tracks the behaviour of latent causal coefficients in all cases. As given in Section 6, in the second inference step we use the mode of $PL$ distribution $\hat{\prec}^0$ to mask the distribution of hidden functions $\mathbf{f}$. Consequently, only those relations that conform to the estimated causal ordering show tight 99% CIs.

## Appendix H  Estimate of $\tau$

In the results below we obtain the estimate $\hat{\tau}$ by means of the estimated GP kernel lengthscale, i.e., $\hat{\tau} = 1/\hat{\lambda}_{\mathrm{RBF}}$. In our experimental assessment, data are generated by using a RBF kernel. Then we can evaluate the goodness of $\hat{\tau}$. However, this assumption might be too restrictive in real-world scenarios, and we might want to use a combination of kernels in the inference procedure. Therefore, the capability of computing $\hat{tau}$ might be impaired since we might not have a single lengthscale resulting from the kernel combination or even we use a kernel that does not involve any lengthscale in its formulation. Having said that, below we provide some insights concerning th estimation of the non-stationarity parameter $\tau$, coming from all the four experimental settings (see Section 7.2).

Figure 14 depicts the inferred values $\hat{\tau}$ for the non-stationarity parameter in the first experimental setting. Here red dashed lines refer to the ground truth value of $\tau$. Our model tends to slightly overestimate the non-stationarity parameter. As the value of $\mu$ increases, the estimate approaches the ground truth.

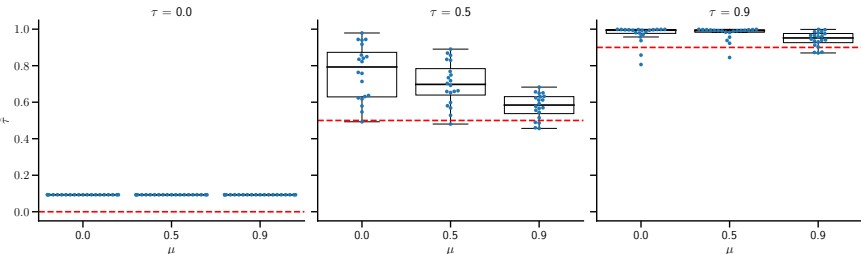

Figure 14: The figure illustrates the box-plots concerning the estimated values $\hat{\tau}$ for the non-stationarity parameter. Red dashed lines refer to the ground truth value of $\tau$.

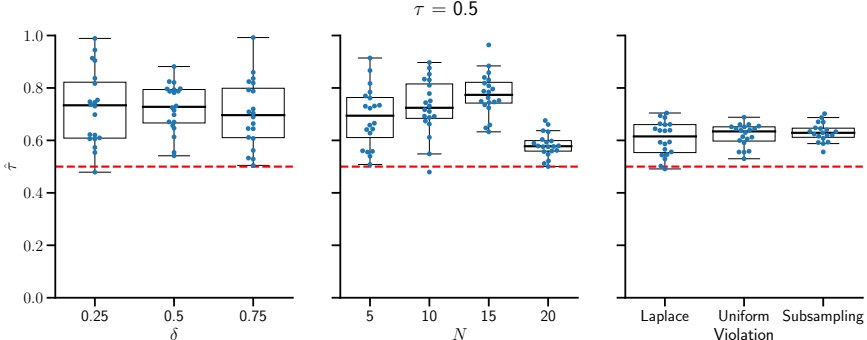

Figure 15: The figure illustrates the box plots concerning the estimated values $\hat{\tau}$ for the non-stationarity parameter. Red dashed lines refer to the ground truth value of $\tau$. On the left, we have the results for the second experimental setting, on the center are those for the third, and on the right are those for the fourth.

In addition, Figure 15 provides the results for the second, third and fourth settings. Overall, we do not appreciate any trend along $\delta$, $N$, and generative model violations. Indeed, MN-CASTLE tends to overestimate the non-stationarity as in the first setting.

## Appendix I    Definitions of the Performance Metrics

In this section we describe the metrics used to evaluate the goodness of the estimated adjacency tensor of the causal graph and the retrieved causal ordering.

**Adjacency.** For the predicted adjacency tensor, we monitor both accuracy and structural scores.

With regards to accuracy measures, we look at the true positive rate ($TPR$, recall), the false discovery rate ($FDR$, 1-precision) and the F1-score. The first is defined as $TP/P$, where $TP$ is the number of predicted edges that exist in the ground truth with the same direction and $P$ (condition positive) is the number of links in the ground truth. The second given by $FP/(FP + TP)$. Here, $FP$ is the number of edges that do not exist in the skeleton of the ground truth, i.e., in the undirected adjacency. Finally, the F1-score is computed as the harmonic mean between $TPR$ and $1 - FDR$ (precision).

Concerning structural metrics, first we consider the *structural Hamming distance* ($SHD$), that represents the number of modifications (added, removed, reversed edges) needed to retrieve the ground truth starting from the estimated network. Then, we also monitor the ratio between the number of predicted edges and the condition positive, given by $NNZ/P$, where $NNZ$ represents the sum of directed ($D$) and undirected ($U$) estimated edges. Finally, we have the fraction of predicted undirected edges, computed as $FU = U/NNZ$.

**Causal Ordering.** To compare the estimated causal ordering with the ground truth, we consider three metrics able to provide a measure of the association strength between two rankings.

First, we look at Kendall-$\tau$, which is a measure of ordinal correspondence between two rankings, bounded between $-1$ (low correspondence) and 1 (strong correspondence). Given two orderings $\hat{\prec}$ and $\prec$, the statistics is defined as:

$$\text{Kendall-}\tau = (P - Q)/\sqrt{((P + Q + T) \cdot (P + Q + U))},$$

where here P is the number of concordant pairs, Q the number of discordant pairs, T the number of ties only in $\hat{\prec}$, and U the number of ties only in $\prec$;

Second, we employ a measure of ranking quality widely applied in information retrieval, the *normalized discounted cumulative gain* (nDCG). Consider a ground truth ordering $\prec$ of length $N$ and suppose to associate items with descending scores $s$, from $N$ to 1. Then, consider an other ordering $\hat{\prec}$ over the same set of elements in $\prec$. Now, define the *discounted cumulative gain* (DCG) as:

$$DCG = \sum_{i=1}^{N} \frac{s_i}{\log_2(i + 1)},$$

and let the *ideal discounted cumulative gain* (IDCG) to be the DCG of $\prec$. Therefore, the nDCG is defined as the ratio by the DCG and the IDCG. This score is bounded between the nDCG of the worst ordering of scores $\bar{s}$, i.e., $s$ sorted in ascending order, and 1. In our analysis we use a min-max scaling to map nDCG to the unit interval. To evaluate the capability of a method in providing high-score items at first positions $k$, we compute the nDCG@k by considering only the first $k$ elements of $\hat{\prec}$.

Finally, we consider Spearman's rank correlation $\rho_S$, that provides a non-parametric correlation coefficient between two series. Here, differently from Pearson correlation, data is not assumed to be normally distributed. Thus, as Kendall-$\tau$, this statistics is bounded between $-1$ and 1. Since in our case we have two score vectors, namely $\hat{\prec}$ and $\prec$, made by distinct values, this metrics can be computed as:

$$\rho_S = 1 - \frac{6 \sum_i (\hat{\prec}_i - \prec_i)}{N(N^2 - 1)}.$$

## Appendix J   Additional Monitored Metrics

In this section we provide additional analysis to better understand the behaviour of the considered methods when (i) we navigate the $(\tau, \mu)$-quadrant keeping the other parameters fixed, (ii) we vary the density of the MN-DAG at a point in the quadrant, (iii) we change the size of the MN-DAG at a point in the quadrant, and (iv) we violate the assumptions of the generative model. Figure 16a, where SHD has been normalized by the number of edges present in the ground truth, refers to the first experimental setting and shows that MN-CASTLE provides the better performances in all settings. Additionally, Figure 16b shows that MN-CASTLE reduces the number of false discoveries returned by baseline models, especially when $\mu \neq 0$. On the contrary, in the single-scale stationary case, best $FDR$ values are provided by GOLEM. Figure 16c tells us that MN-CASTLE is the best in the retrieval of true positives in almost all cases. Furthermore, the estimated to true edge set ratio given in Figure 16d shows that our model is the more accurate in the number of causal connections. Finally, we plot the fractions of undirected connections given by our model to check aciclicity of the retrieved structure.

With regards to causal ordering estimation, Figure 17 provides results also for Kendall-$\tau$ (Figure 17a), Spearman's rank (Figure 17b) and nDCG@5 (Figure 17c) statistics. Here, the methodology used is the same as in Section 7.2. According to these metrics, MN-CASTLE outperforms the random baseline model in all cases. In addition, the values of the monitored metrics do not lower as $\tau$ grows and improve as $\mu$ increases.

Hence, Figures 18 to 20 depict the resulting values for the same metrics above, obtained in the second, third, and fourth experimental settings, respectively. Overall, when the density of the network grows, the metrics improve for all methods. Also, MN-CASTLE provides the best performance for all values of $\delta$. When we vary the network size, our method outperforms the baselines as well. The overall worsening in method's performance is due to the fact that here the dimensionality of the problem grows, while the number of observations is kept fixed, $T = 100$. In addition, the performance of our method remains good even under generative model misspecification.

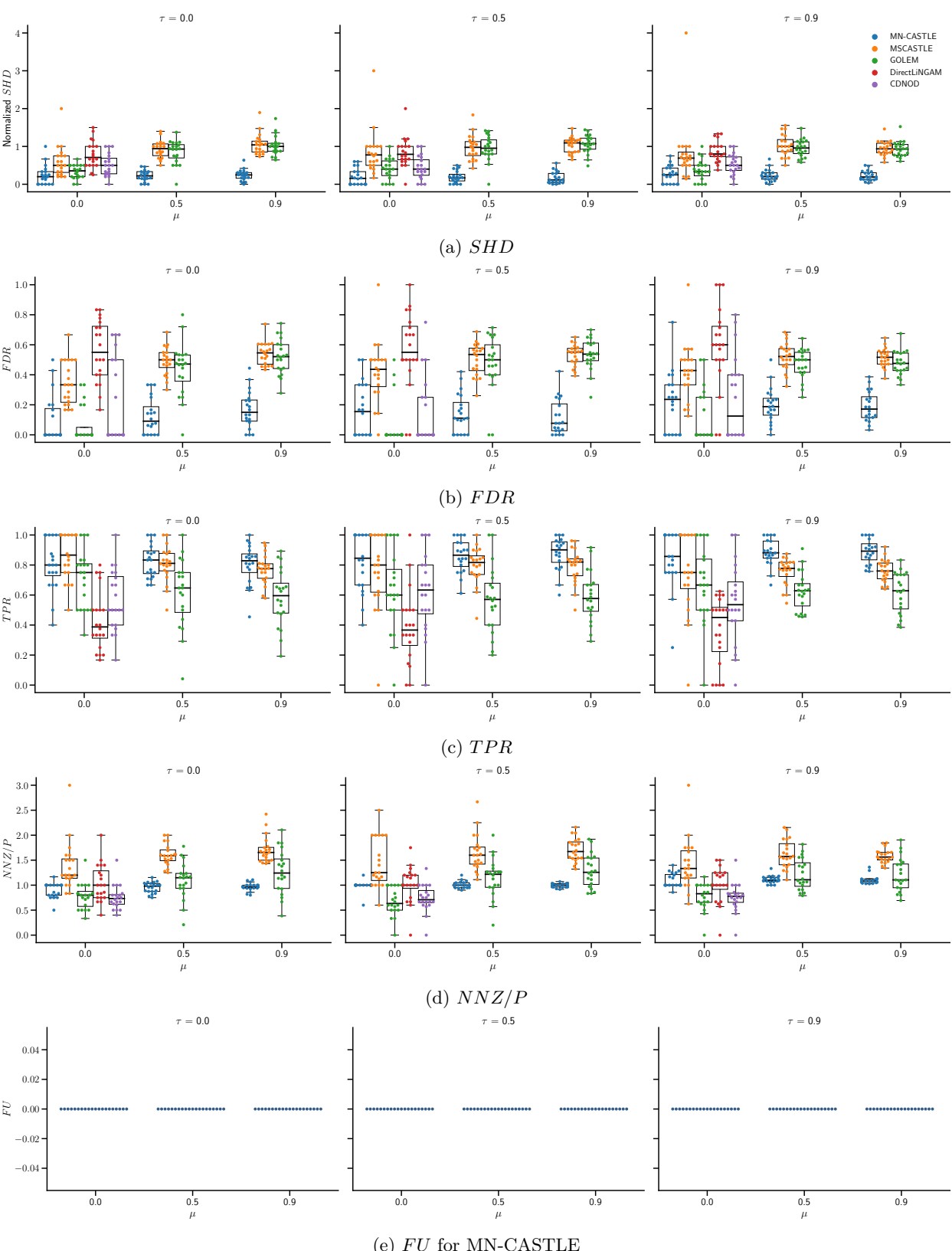

Figure 16: The figure depicts results returned by additional monitored metrics for the considered $(\tau, \mu)$ settings. Here we use box plots, where we overlay the values of the metric, plotted as points.

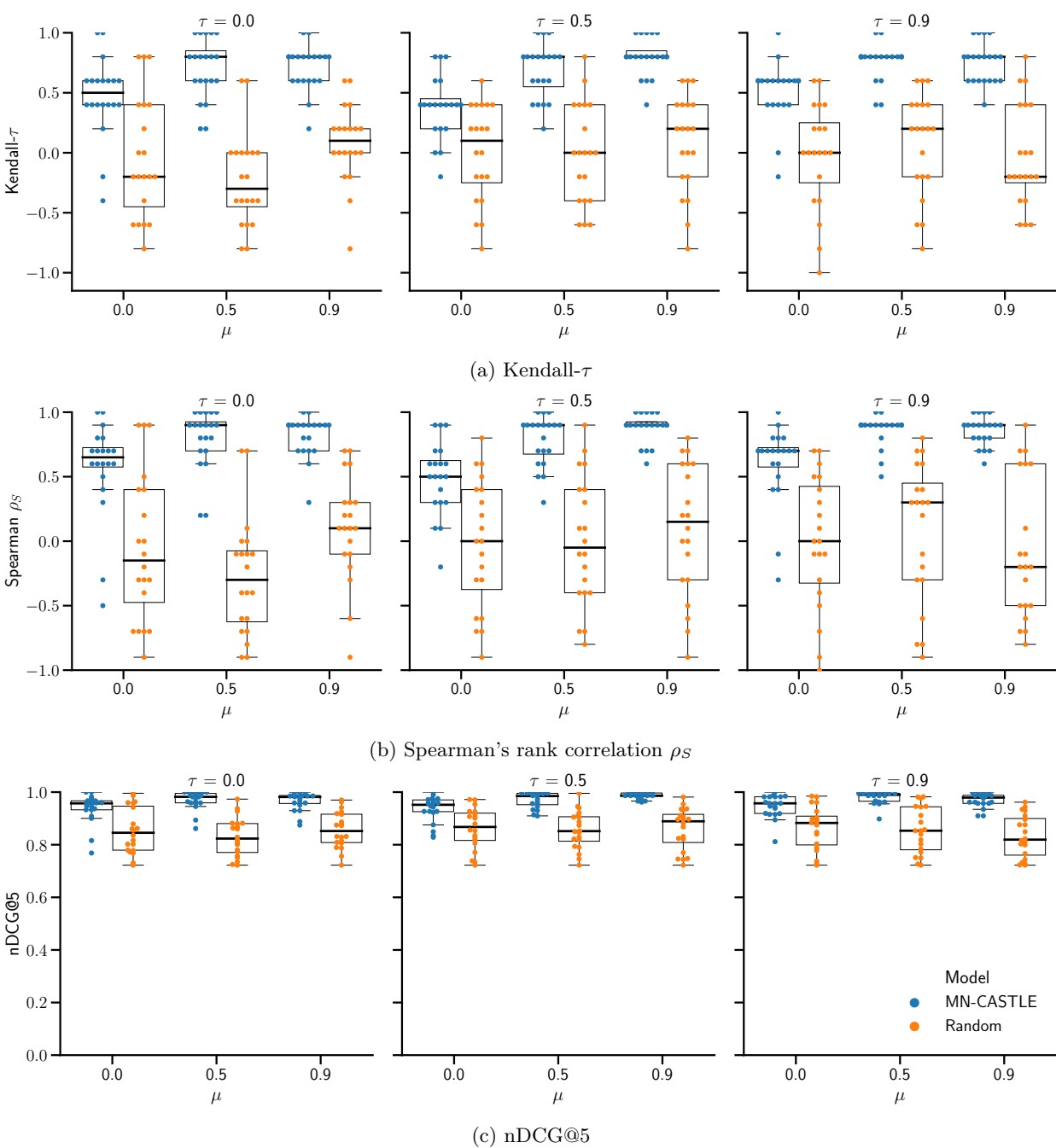

(a) Kendall-$\tau$

(b) Spearman's rank correlation $\rho_S$

(c) nDCG@5

Figure 17: The figure depicts box plots along with quartiles reference lines (dashed lines) for (a) for Kendall-$\tau$ metric, (b) Spearman's rank correlation, and (c) normalized discounted cumulative gain (nDCG) at 5. MN-CASTLE is given in blue while a random baseline model in orange. For every dataset generated according to a given $(\tau, \mu)$ setting (i) we sample $1 \times 10^3$ causal orderings $\hat{\prec} \sim PL(\hat{\boldsymbol{\theta}})$, where $\hat{\boldsymbol{\theta}}$ is the estimated vector of scores; (ii) we draw $1 \times 10^3$ random causal orderings $\bar{\prec} \sim PL(\bar{\boldsymbol{\theta}})$, where $\bar{\theta}_i \sim U(0, N)$, $i = 1, \ldots, N$. Afterwards, we evaluate the three monitored metrics by using the sampled causal orderings and $\prec$ for both models. Thus, each box plot is made by $2 \times 10^4$ points.

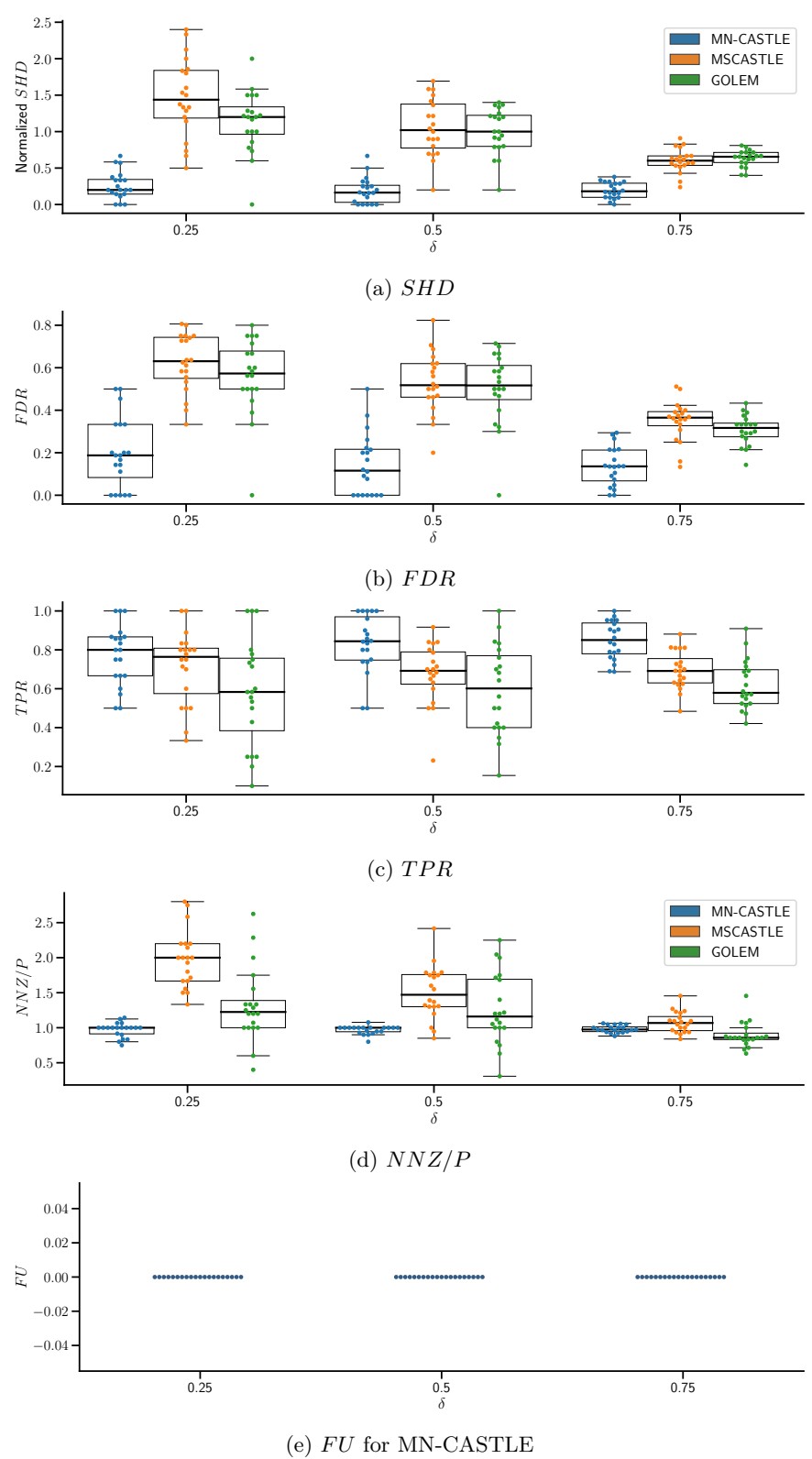

(a) $SHD$

(b) $FDR$

(c) $TPR$

(d) $NNZ/P$

(e) $FU$ for MN-CASTLE

Figure 18: The figure depicts results returned by additional monitored metrics for different MN-DAG densities $\delta$. Here we use box plots, where we overlay the values of the metrics, plotted as points.

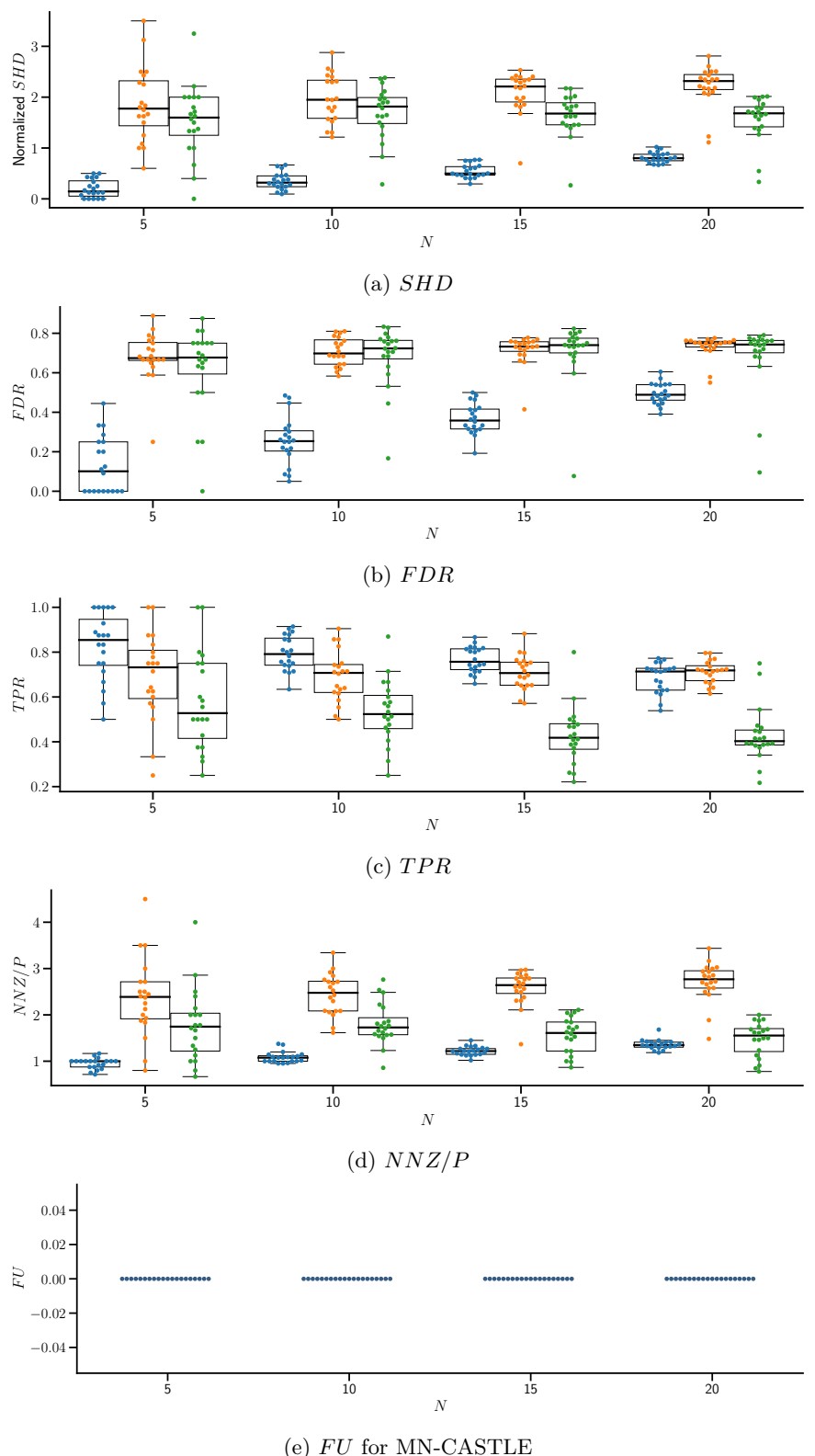

(a) *SHD*

(b) *FDR*

(c) *TPR*

(d) *NNZ/P*

(e) *FU* for MN-CASTLE

Figure 19: The figure depicts results returned by additional monitored metrics for different number of nodes *N*. Here we use box plots, where we overlay the values of the metrics, plotted as points.

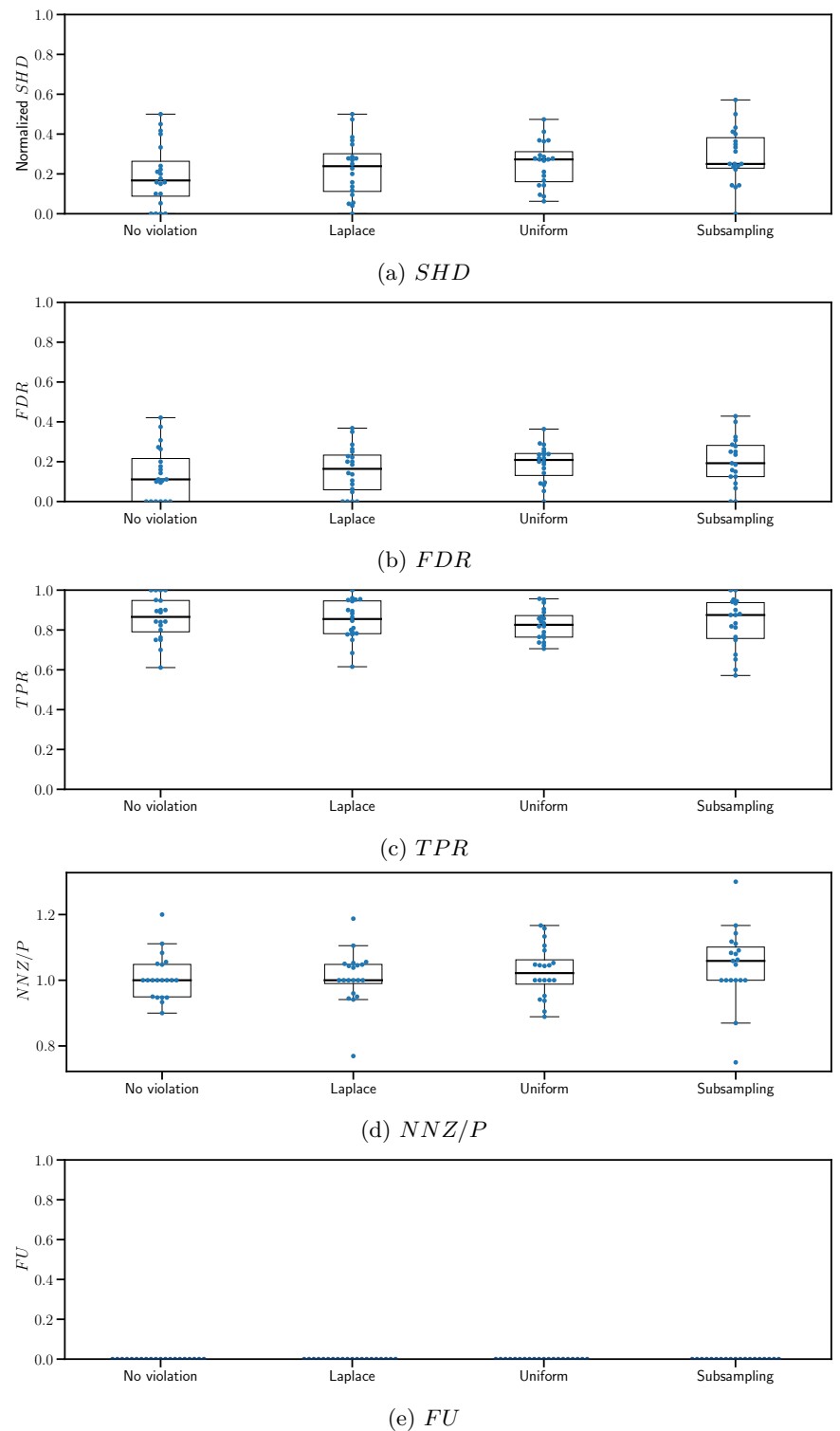

(a) *SHD*

(b) *FDR*

(c) *TPR*

(d) *NNZ/P*

(e) *FU*

Figure 20: The figure depicts results returned by additional monitored metrics for different kind of generative model misspecification. Here we use box plots, where we overlay the values of the metrics, plotted as points. To ease the comparison, we also provide the case without any violation.

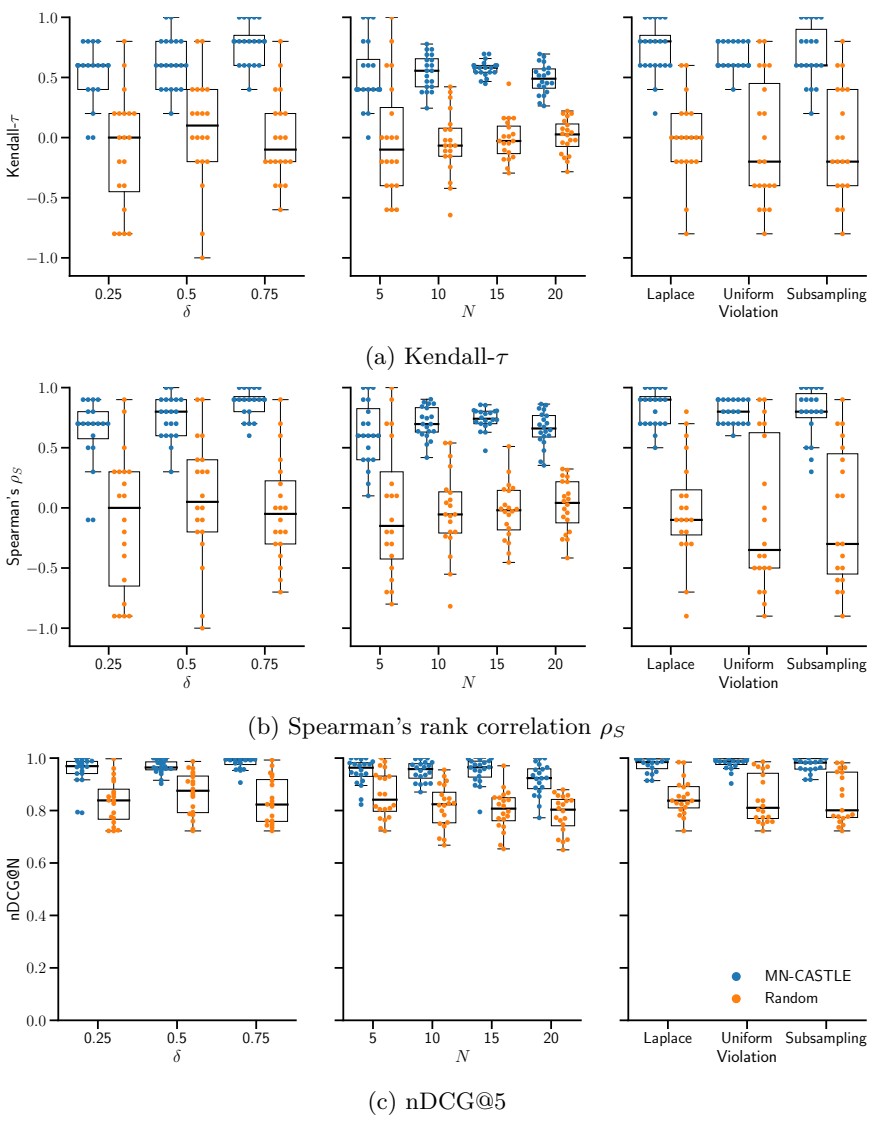

(a) Kendall-$\tau$

(b) Spearman's rank correlation $\rho_S$

(c) nDCG@5

Figure 21: The figure depicts box plots for (a) Kendall-$\tau$, (b) Spearman's rank correlation, and (c) the normalized discounted cumulative gain (nDCG) at $N$. MN-CASTLE is given in blue while a random baseline model in orange. Box plots on the left refer to the second experimental configuration, i.e., when we vary $\delta$ while keeping fixed the values of the others parameters, as described in Section 7.2. Box plots on the center concern the third experimental setting, where we study the sensitivity of the estimation accuracy w.r.t. the network size $N$. Box plots on the right relate the fourth experimental setting, where we study MN-CASTLE under generative model misspecification. Notice that in the left and right plots $N = 5$.

Finally, Figure 21 depicts the results for Kendall-$\tau$ statistics, the Spearman's rank correlation, and the nDCG at $N$ related to causal ordering estimation. Here, we see that MN-CASTLE performance grows along with $\delta$, does not show dependence (in mean terms) on the size of the underlying MN-DAG, and does not decrease under generative model violation.

## Appendix K   Data sources, Pre-processing and Inference Details

We obtained data on a weekly basis from January 1, 2018 to December 31, 2022. In detail, we downloaded Henry Hub natural gas futures prices (NG), WTI futures prices for crude oil (CO), New York Harbor No. 2 Heating Oil futures prices (HO), and US natural gas storage (ST) data from the website of the US Energy Information Administration (EIA). The crack spread was calculated as the difference between HO and CO, with HO being converted to dollars per barrel. The deviation of storage from the norm (SDD) was determined by comparing the ST value for a given week to the average value for the same week over the previous five years. We also downloaded rig counts (RC) data from Baker Hughes, and extracted deviations from seasonal average values of gas consumption for cooling (CDD) and heating (HDD) environments from the National Oceanic and Atmospheric Administration website. Finally, the economic uncertainty index (UI) was downloaded from the Federal Reserve Economic Data repository.

We have checked for non-stationarity in the time series using the ADF test at a 99% level of significance. We have found evidence of a unit root in all time series except CDD and HDD. Therefore, we have taken the first differences of the non-stationary time series. Finally, to account for differences in scale among the time series, we have standardized each time series.

After the preprocessing, we have obtained an estimate of $\mathbf{S}_j$ by using the R package `mvLSW` ((Taylor et al., 2019)). The wavelet transform has been performed using Daubechies wavelet with filter length equal to 8. In addition, the smoothing of the periodogram has been performed using the rectangular kernel, also known Daniell window, of width 16. Finally the smoothed periodogram has been corrected for the bias by using the inverted autocorrelation wavelet ((Eckley & Nason, 2005)) and regularized to ensure positive definiteness ((Schnabel & Eskow, 1999)). At this point, we have obtained an estimate of $\mathbf{O}_j$ by means of inversion for the first 5 finer scales. Indeed, the values for the estimate $\widehat{\mathbf{S}}_j$ were not negligible over these 5 scales.

In order to estimate the mode of the PL distribution $\prec^0$, we have proceeded as follows. We have run the first step of the inference procedure $n = 1000$ times, and we have obtained $n$ estimates $\hat{\boldsymbol{\theta}}_i$, $i \in [n]$. Then, we have computed the estimate $\hat{\boldsymbol{\theta}}$ by calculating the median over the previous $n$ estimates. At this point, we have obtained $\prec^0 = \mathrm{argsort}(-\hat{\boldsymbol{\theta}})$. Indeed, since our model is probabilistic, we may have obtained different estimates of $\hat{\boldsymbol{\theta}}$ at each run.

Finally, to infer the causal coefficients we have used a combination of kernels, specifically $K = K_{\mathrm{Periodic}} + K_{\mathrm{Linear}} \times K_{\mathrm{Matern}^{3/2}}$ and we have set $\rho = 0.05$ for hard-thresholding the partial coherence (see Section 6).

## Appendix L   Complete MN-DAG of the Real-world Application

Figure 22: The figure depicts the multiscale DAG retrieved by MN-CASTLE, where nodes are sorted according to the estimate $\prec^0$.

