# OpenReview forum: "Learning Multiscale Non-stationary Causal Structures"
_TMLR — Accepted by TMLR_

### Review · Reviewer_Luq6 · 2023-05-26

**Summary Of Contributions:**

The paper studies the problem of causal structure learning from time-series data. This paper introduces multiscale non-stationary directed acyclic graphs (MN-DAGs) which generalizes linear DAGs to the time-frequency domain. The authors  develop a probabilistic generative model that allows sampling MN-DAGs based on user-specified priors on the time-dependence and multiscale properties of the causal graph. A Bayesian method (called MN-CASTLE) for learning the MN-DAGs is further proposed by using stochastic variational inference. Experiment results on synthetic and real world data are provided.

**Audience:**

Yes

**Claims And Evidence:**

Yes

**Requested Changes:**

As mentioned, my major concerns have been addressed by the resubmitted manuscript.

**Strengths And Weaknesses:**

Strength:
- The paper is well written and clear.
- The topic considered in this work is interesting and important because it is common to encounter non-stationary in real world.
- The proposed model and estimation method seem to be sound to my knowledge.
-  Experiments are conducted across different settings and the results are convincing.

Weakness: My major concerns have been addressed by the resubmitted manuscript. There was some concern related to identifiability of the proposed model, which the paper discussed as part of future research directions in Section 8.

---

> ### Author Response · Authors · 2023-05-26
> **Reply to Reviewer's comment**
>
> We would like to thank the Reviewer once again for their helpful comments, which have greatly contributed to the improvement of our work, and for their kind appreciation.

---

### Review · Reviewer_URns · 2023-06-18

**Summary Of Contributions:**

The paper proposes MN-DAG, which extends structural causal models to include nonstationarity and a multiscale property (causal relations changing over time).

A generative model is describe which involves (i) sampling the number of time scales from a binomial distribution, (ii) sampling a causal ordering according to the Placket-Luce distribution for each time scale, (iii) sampling and masking to generate weight matrices. Data may then be sampled from this model using the locally stationary wavelet process.

The authors then describe MN-CASTLE, an algorithm for learning MN-DAGs from such data using stochastic variational inference.

Experiments show data sampled from the MN-DAGs have the expected properties, MN-CASTLE is more effective at learning from synthetic data simulated from using the described procedure when the multiscale property is high and MN-CASTLE is able to estimate multiscale causal structures in the US natural gas market that vary at the beginnings of the COVID-19 pandemic and Ukraine war.

**Audience:**

Yes

**Claims And Evidence:**

Yes

**Requested Changes:**

This submission represents an improvement both in terms of novelty (Lemma 4.2 and Proposition 4.3) and experimental results (overall improvement and the inclusion of the US natural gas market example which exhibits the multiscale property in real world data), which were my primary two concerns when reviewing an earlier version of this paper.

I would still, however, recommend that in the introduction the authors replace 'structural equation models' with 'additive noise-based' or something similar to distinguish ANM / LiNGAM approaches from constraint-based and score-based structure learning methods, since the later two refer to structure learning and 'structural equation models' is typically used to refer to the framework more generally (not a specific learning procedure).

**Strengths And Weaknesses:**

Strengths
- The model is ambitious and general in permitting both nonstationarity and the multiscale property
- Extensive theoretical background is included
- Real world experimental results demonstrate the effectiveness of the method

Weakness
- While the method contains some novel improvements (which improves from the previous submission), the generative model and learning procedure are based on off the shelf methods and frameworks so novel content is somewhat limited

---

> ### Author Response · Authors · 2023-06-23
> **Reply to Reviewer's comment**
>
> We would like to thank the Reviewer once again for their helpful comments, which have allowed us to improve our work. Additionally, we believe that there is room for further improvement in the current usage of the term _"structural equation models"_, as suggested by the Reviewer.
>
> Specifically, while we consider the current reference to _"structural equation models"_ in Sec. 1 appropriate (as we are referring to an approach rather than a specific learning procedure), in Sec. 2, we will replace the term with _"functional causal models"_ and include a reference to the post-nonlinear model [1] to encompass models where a functional form is applied to an additive noise model.
>
> \
> References\
> [1] Zhang, Kun, and Aapo Hyvarinen. "On the identifiability of the post-nonlinear causal model." arXiv preprint arXiv:1205.2599 (2012).

---

### Review · Reviewer_KzQ3 · 2023-09-07

**Summary Of Contributions:**

The paper proposes a graphical model, MN-DAG, to multiscale and nonstationary time series datasets. Generation of MN-DAG uses hyperparameters to model different time scale and nonstationarity among data. An variational Bayes method to infer the graph structure is provided, based on causal order and MN-DAG. Experimental results show promising performances of the proposed methods over a set of baselines.

**Audience:**

Yes

**Claims And Evidence:**

Yes

**Requested Changes:**

Besides the question above, the organization of the paper should be more carefully thought. Before talking about Sampling an MN-DAG, one section should discuss the model itself (instead of in the introduction). The order of data generation and then learning also seems confusing.

**Strengths And Weaknesses:**

Strength:

- The paper discusses both the generation and learning of the MN-DAG.
- the problem is not carefully studied before, and hence a good novelty
- proposed method is non-trivial, although most are based on components of some existing DAG approaches.

Weakness:

- No guarantees on the soundness or consistency of the learned graphs.


Some comments/questions:
1. time scale 2^j: does it mean there are two graphs for j = 1, 4 for j=2 etc?
2. "Causal inference methods": typically these are called causal discovery methods to infer graphs, inference are typically limited to parameter and ATE as such.
3. Section 3 first paragraph: seems like an exact repeat of a paragraph in the introduction.
4. Proposition 4.3: How does one understand the spectral properties of X_T? is it the graph property?
5. Figure 4: caption
6. Does the policy gradient based learning method require a lot of data samples?
7. It was not fully clear to me how the hyperparameters $\mu$ and $\tau$ are used in the experiments. Appendix H should be the main text of the paper on estimation, and does not discuss $\mu$.  Are $\mu$ given or estimated?

---

> ### Author Response · Authors · 2023-09-20
> **Response from the Authors [Part 1]**
>
> **Strength**\
> **[S]** We sincerely thank the Reviewer for recognizing the novelty of our work and for their appreciation.
>
> **Weakness**\
> **[W]** We agree with the Reviewer that it would be desirable to have some theoretical guarantees. However, rather than being a weakness of our approach, this is an intrinsic limitation associated with causal structure learning because, in real-world contexts, the ground truth is unknown and in general it is not possible to guarantee its exact recovery (see, e.g., Proposition 7.1 in [1]). Moreover, in real-world contexts, identifiability guarantees could be easily violated (e.g., due to the presence of unmeasured confounders, measurement errors, or limited data availability), impairing the performance of causal discovery methods [2, 3, 4].
>
> We thank the Reviewer for their questions and comments, that we address below. In our response, we will refer to the uploaded revised version of the article.
>
> **[Q1]** No, each scale level $j$ is associated with one graph, representing the causal structure at the corresponding temporal resolution.
> To better clarify the meaning of temporal scales, let us consider a data set $\mathbf{X} \in \mathcal{R}^{N \times T}$ made by $N$ time series of length $T=2^J$, with $J \in \mathcal{N}$. Each column $\mathbf{x}[t]=[x_1[t], \ldots, x_N[t]]^T$ represents a sample collected at frequency $\Delta t$. For example, let’s say that the time series are observed at daily frequency, hence $\Delta t$ is equal to one day.
> The scale level $j=1$ refers to the variations of the time series associated with temporal resolution of $2^1 \Delta t$, i.e., two consecutive days. Analogously, the scale level $j=2$ refers to the variations of the time series associated with temporal resolution of $2^2 \Delta t$, i.e., four consecutive days. And so on and so forth, until we reach the maximum level $j=J$. For a more comprehensive description, we refer the Reviewer to Sec.2 of [5] and references therein.\
> To improve the clarity of our paper, we have included these details in Sec.1 on lines 47-54.
>
> **[Q2]** We thank the Reviewer for this very helpful comment. Since in the submitted version of the article, we frequently used the term *causal structure learning* (which we think describes more clearly the task), in the revised version, we have replaced *causal inference* with *causal structure learning* throughout the paper.
>
> **[Q3]** As per the Reviewer’s requested change, in the revised version we have removed the overlap.
>
> **[Q4]** As indicated on line 236, given the multivariate time series $\mathbf{X}_T$ generated according to Equation (2), when we say spectral properties we refer to (i) the local wavelet spectral matrix $\mathbf{S}_j[\nu]$, describing the cross-covariance between the time series at a certain scale level $j$ and time $\nu$; and (ii) the inverse of $\mathbf{S}_j[\nu]$, namely $\mathbf{O}_j[\nu]$ that encompasses the information regarding the conditional linear dependence between two time series.\
> Preposition 5.3 proves that if you permute (i) the causal structure (involved in Equation (2) via the definition of $\mathbf{M}$) or (ii) the order of the time series within $\mathbf{X}_T$, the spectral properties are permuted accordingly, meaning that the cross-covariance and the conditional linear dependence between two time series remain the same after the permutation.
>
> **[Q5]** We have corrected the caption in the revised version, correctly positioning the reference to Fig. 4a. The same modification has also been made to Fig. 3.
>
> **[Q6]** In our experiments, we studied the performance of the proposed method while varying $N$, with $T$ fixed (see Fig. 8b). This analysis also provides insights into the number of samples required by the method to achieve good performance. In particular, we observe that when $T \geq N^2$, the F1 score exceeds the value of $0.8$.
>
> **[Q7]** The parameter $\mu$ determines in a probabilistic way the number of time scales at which causal relationships show up, namely $J$, as described in the paragraph *“Sample the time scales”* in Sec.4. In the experiments, $\mu$ neither is given as input nor is estimated by the proposed method. Our learning method takes as input the matrices $\widehat{\boldsymbol{\Omega}}_j$, $j \in [J]$, as described in Sec.6 and depicted in Fig.4. Hence, the method assumes that the causal relations could occur only at the provided scale levels $j \in [J]$.\
> Since our method operates in a single-domain setting (i.e., it focuses on a single $J$ depending on the binomial distribution parameterized by $\mu$), it has no possibility of estimating $\mu$. The extension of the proposed method to a multi-domain setting, where it might potentially be possible to estimate $\mu$, is an intriguing research direction.

---

> > ### Author Response · Authors · 2023-09-20
> > **Response from the Authors [Part 2]**
> >
> > **Requested change**\
> > **[RC]** We thank the Reviewer for the suggestion. In the revised version, we present the probabilistic generative model in Sec.3, right before the section about sampling an MN-DAG. \
> > As for the second request, we have improved the "Roadmap" in Sec.1 (lines 89-93), so as to dispel any doubts regarding the reasons for addressing data generation in Sec.5 and the learning method in Sec.6. Specifically, in addition to defining the data generative process, Sec.5 provides key insights into the relationship between the causal structure and the spectral properties of the generated time series. These results, formalized in Lemma 5.2 and Proposition 5.3, rigorously support and justify the two-step inference procedure described in Sec.6. Additionally, they are instrumental in explaining and understanding the choice of observables in the probabilistic models depicted in Fig. 3 and Fig. 4 in Sec.6. Therefore, we have consciously described the learning procedure after the data generation part, as the proposed learning method is shaped by the results presented in Sec.5.
> >
> > **References:**
> >
> > [1] Peters, Jonas, Dominik Janzing, and Bernhard Schölkopf. Elements of causal inference: foundations and learning algorithms. The MIT Press, 2017.\
> > [2] Zhang, Kun, et al. "Causal discovery in the presence of measurement error: Identifiability conditions." arXiv preprint arXiv:1706.03768 (2017).\
> > [3] Gao, Erdun, et al. "MissDAG: Causal discovery in the presence of missing data with continuous additive noise models." Advances in Neural Information Processing Systems 35 (2022): 5024-5038.\
> > [4] Kaltenpoth, David, and Jilles Vreeken. "Causal Discovery with Hidden Confounders using the Algorithmic Markov Condition." Uncertainty in Artificial Intelligence. PMLR, 2023.\
> > [5] Percival, Donald B., and Harold O. Mofjeld. "Analysis of subtidal coastal sea level fluctuations using wavelets." Journal of the American Statistical Association 92.439 (1997): 868-880.

---

### Author Response · Authors · 2023-09-20
**We have uploaded a revised version of our paper addressing the comments of the reviewers.**

Dear AE and Reviewers,

Thank you for the time spent in managing and reviewing our work. We have uploaded a revised version of the paper, including the requested changes from the reviewers.

Specifically, we have streamlined Sec.1, by moving the presentation of the probabilistic generative model in a dedicated section, i.e., Sec.3. Additionally, we have corrected the usage of the term “structural equation models”.

To facilitate the review of the modifications, we have used the following visual aids:
1. New content in the manuscript is highlighted in blue.
2. Existing content that has been relocated within the text is marked in red.
3. Line numbers have been added for reference.

Sincerely,\
The Authors.

---

### Decision · Action_Editor_qS4v · 2023-10-26

**Recommendation:** Accept as is

**Comment:**

I apologize for the delay in this decision. The reviewers and I are in agreement that this paper should be accepted to TMLR, and I recommend accepting as-is.

The paper has improved dramatically from its original submission, and the reviewers remaining concerns about this version have been addressed by the authors. In particular, the inference methodology has been improved and the experimental section expanded and made more realistic. The updated paper is well written, of interest to the community, and describes novel and relevant work.

**Audience:**

Yes, nonstationary time series data is of broad interest, and a significant proportion of TMLRs audience would be interested in causal methods for such data.

**Claims And Evidence:**

The methodology is supported by compelling experimental analysis. The experiments are much improved from the original submission.

---

> ### Author Response · Authors · 2023-10-30
> **Thank you, we have uploaded the camera-ready version of the manuscript.**
>
> Dear AE, dear Reviewers,
>
> We sincerely thank you for your work in managing and reviewing our submission. We are glad to receive the news that our manuscript has been accepted for publication in Transactions on Machine Learning Research.
>
> We have uploaded the camera-ready version of the manuscript.
> In particular, we have replaced the old title with a shorter one, which we also believe fits the journal's template better.
>
> Yours sincerely,\
> The Authors.